# TRIDENT: An Efficient Data-Free Model Extraction Attack for Graph Neural Networks

## Abstract

Graph Neural Networks (GNNs) are increasingly offered as Machine-Learning-as-a-Service (MLaaS) via APIs. However, this deployment model exposes them to model extraction attacks (MEAs), where adversaries aim to reconstruct or steal the proprietary GNN models by leveraging only query access to the service. Although recent work has developed data-free MEAs that have performed well in both transductive and inductive settings, prior efforts on GNN extraction have been largely empirical and lacked a theory-driven blueprint. In this paper, we formalize extraction risk and derive a lower bound that ties it to the victim's own risk and the generalization discrepancy between the query and target distributions, yielding a clear design principle: effective attacks must minimize discrepancy. Guided by this, we propose TRIDENT (daTa-free queRy-efficIent moDel ExtractioN aTtack), which strategically schedules queries, particularly under strict budget constraints. Extensive experiments on six real-world benchmarks and three GNN backbones show that our method achieves state-of-the-art performance. These results validate both the theoretical contributions and the practical efficiency of our approach.

## 1 Introduction

Graph Neural Networks (GNNs) Scarselli et al. (2008); Kipf & Welling (2016); Hamilton et al. (2017) have shown remarkable success in modeling graph-structured data, achieving promising performance in a wide range of applications such as recommendation systems Ying et al. (2018); Gao et al. (2022), molecular property prediction Gilmer et al. (2017); Wieder et al. (2020) and fraud detection Dou et al. (2020); Liu et al. (2021). To facilitate their deployment at scale, Graph-based Machine Learning as a Service (GMLaaS) Ribeiro et al. (2015); Zhang et al. (2020) has emerged as a popular paradigm, enabling users to access powerful GNN models via cloud-based APIs. This paradigm has accelerated the adoption of GNNs in critical domains such as healthcare Paul et al. (2024), e-commerce Zhang et al. (2022a), and scientific research Zhang et al. (2021a); Thais et al. (2022). However, GMLaaS also introduces security vulnerabilities Guan et al. (2024), among which model extraction attacks (MEAs) Tramèr et al. (2016); Orekondy et al. (2019); Dziedzic et al. (2022); Jagielski et al. (2020); Liang et al. (2024) pose a particularly significant threat. Adversaries may attempt to replicate a target model by issuing queries and collecting output responses. In contrast to adversarial perturbations Finlayson et al. (2019); Guo et al. (2019) that aim to degrade model performance, MEAs focus on stealing the GNN models themselves, which may be proprietary or critical intellectual properties of the model owners Yu et al. (2020); Oliynyk et al. (2023), allowing attackers to recreate these models without the cost of data collection or training. This not only undermines commercial interests but also leads to secondary privacy breaches if the surrogate model is exploited for further attacks.

Early explorations of MEAs against GNNs were largely empirical DeFazio & Ramesh (2019); Wu et al. (2022), lacking foundational theoretical analysis. Recent advancements Shen et al. (2022), including sophisticated data-free MEAs like Zhuang et al. (2024), have emerged. However, a deep theoretical understanding, especially under practical constraints (e.g., limited queries, black-box models, no training data), is still notably scarce Xian et al. (2022). This theoretical gap is not confined to GNNs; even for general MEAs, rigorous theoretical backing is still limited. This widespread deficiency motivates our central question: *what kinds of queries enable effective MEAs?* Answering this necessitates developing a generalized theoretical approach to MEAs against GNNs.

To address this gap, we aim to develop a theory-based framework for modeling MEAs against GNNs in strictly real-world settings. Specifically, we consider an adversary who has access to a pre-trained model deployed via GMLaaS, but can only interact with it through a limited number of queries. The adversary's goal is to construct a surrogate model that effectively captures the decision boundary of the target model while operating under constraints on time, memory, and query budgets. Addressing this question requires more than algorithmic development—it calls for a generalizable theoretical model of MEAs. Yet, formulating such a theory-based framework is non-trivial due to several fundamental obstacles inherent to the data-free, query-limited setting: (1) **No access to the training data.** In real-world GMLaaS scenarios, attackers often face a *data-free* environment where they have no access to the original training data or similar datasets is available. This may lead to severe distribution shifts, as synthetic queries may deviate significantly from the target model's training distribution, causing poor surrogate model performance. (2) **Limited Query Budgets.** Service providers routinely throttle API usage, imposing tight budgets on the number of allowable queries per time window, which forces a careful balance between *exploration* (understanding the decision boundary) and *exploitation* (refining surrogate performance), making query efficiency critical. (3) **Lack of theoretical modeling.** Existing theoretical analysis of MEAs Xian et al. (2022); Karmakar & Basu (2023); Zhang et al. (2025) concentrates on simplified settings, which cannot explain MEA performance under practical constraints.

To address these challenges, we propose TRIDENT (da$\underline{T}$a-free que$\underline{R}$y-effic$\underline{I}$ent mo$\underline{D}$el $\underline{E}$xtractio$\underline{N}$ a$\underline{T}$tack), a theory-driven MEA on GNNs. Specifically, to tackle the first challenge, we extend the idea from data-free model extraction Truong et al. (2021) and design a budget ratio to sacrifice queries in each iteration to generate more graphs. Next, to address the second challenge, we propose Surrogate-as-Victim Alignment to train a light surrogate model through MEAs targeting the surrogate model, providing a white-box setting. By integrating the KL divergence of the two surrogate models' output into the loss, the surrogate model potentially focuses more on uncertain nodes, since the light version possesses some properties of the surrogate model. This introduces a high degree of alignment, while no extra queries are needed. Finally, to overcome the third challenge, we propose a *Unified Theoretical Framework* that bridges the gap between empirical attacks and theoretical understanding, offering explainability. Our main contributions can be summarized as follows:

- **Theoretical Analysis:** We provide a comprehensive *theoretical analysis*, involving deriving bounds that connect extraction risk with generalization discrepancy, thereby guiding the development of more effective and practical attack strategies against GNNs.

- **Theory-to-design principle:** We claim the first theory-driven framework for data-free GNN extraction: earlier GNN-extraction work was mainly empirical, whereas we formalize extraction risk in this context and use it to derive a guiding principle for attack design.

- **Experimental Evaluation:** Extensive experiments across multiple datasets and GNN architectures demonstrate that our method, TRIDENT, outperforms state-of-the-art baselines (e.g., STEALGNN Zhuang et al. (2024)) in both fidelity and efficiency, even under defense mechanisms.

## 2 PROBLEM FORMULATION

**Notations.** Let $X, Y$ be jointly distributed under a probability measure $\mu$, where $X \in \mathcal{X} \subset \mathbb{R}^n, Y \in \mathcal{Y} \subset \mathbb{R}$ represent the features and labels respectively. In classification tasks, $Y$ is a finite set. A hypothesis class $\mathcal{H}$ consists of measurable functions $\mathcal{H} = \{h \in \mathcal{H}|h : \mathcal{X} \mapsto \mathcal{Y}\}$ with a loss function $\ell : \mathcal{Y}^2 \mapsto \mathbb{R}_+$. Given a distribution $\mu$, the expected risk is defined as: $R_\mu(h) := \mathbb{E}_\mu[\ell(h(X), Y)]$. We provide a comprehensive notation table in Appendix B.

To formally establish the problem of model extraction, particularly in the context of GMLaaS, where attackers interact with a victim graph learning model, we introduce the general background first. Real-world attackers do *not* observe the training distribution of a victim GNN, yet they evaluate extraction quality against the victim's *outputs*. This scenario necessitates the introduction of three key notions: (i) *extraction risk* (Def 1): to quantify the loss between a surrogate model and the victim GNN's outputs on queried data; (ii) *generalization discrepancy* (Def 2, Def 3): to measure how any model's risk, including GNNs, changes when evaluated on the attacker's query distribution versus the original distribution; (iii) *incremental extraction* (Def 6): to formalize the multi-round query-and-train process an attacker might employ against the GNN. These concepts are formalized below.

**Definition 1 (Extraction Risk)** *Given a fixed function $h \in \mathcal{H}$ (e.g., a surrogate GNN) and a reference model $h_{\mu'}$ (e.g., the victim GNN), the extraction risk of $h$ with respect to a distribution $\mu$ (over graph data) is defined as $\tilde{R}_\mu(h) := \mathbb{E}_\mu[\ell(h(X), h_{\mu'}(X))]$.*

In the context of model extraction attacks, we consider a target distribution $T$ (on which the victim GNN is trained) and a query distribution $Q$ (from which the attacker draws graph samples to query the victim GNN), both defined over the joint space $(X, Y)$. The target model, a GNN in our scenario, is denoted $h_T : \mathcal{X} \mapsto \mathcal{Y}$. For clarity in the subsequent discussion, we will replace $\mu, \mu'$ with $Q, T$ unless otherwise specified throughout this paper. Throughout this paper, the input space $\mathcal{X}$ is instantiated for graph-structured data, and $\mathcal{X}$ can be naturally interpreted as the product measurable space $\mathcal{A} \times \mathbb{R}^{N \times d}$.

To evaluate the surrogate on unseen data, we additionally define the test risk, given by $\tilde{R}_{T'}(h) := \mathbb{E}_{T'}[\ell(h(X), h_T(X))]$, which reflects how well $h$ approximates $h_T$ on the test distribution $T'$.

**Definition 2 (Generalization Discrepancy)** *For any $h \in \mathcal{H}$, the generalization discrepancy between two distributions $\mu'$ and $\mu$ is defined as $\Delta_h(\mu, \mu') := R_{\mu'}(h) - R_\mu(h)$.*

**Definition 3 (Extraction Generalization Discrepancy)** *For any $h \in \mathcal{H}$, the extraction generalization discrepancy is defined as $\tilde{\Delta}_h(\mu, \mu') := \tilde{R}_{\mu'}(h) - \tilde{R}_\mu(h)$.*

Let $\{(x_{T,i}, y_{T,i})\}_{i=1}^{n_T}$ denote the i.i.d. samples (graphs and their labels) used to train the target model $h_T$, drawn from the target distribution $T$. Similarly, query graphs $\{x_{Q,j}\}_{j=1}^{n_Q}$ are drawn i.i.d. from the query distribution $Q$. Note that while the training and querying distributions may differ, i.i.d. sampling is assumed within each dataset. We define a family of querying strategies $f_A \in \mathcal{F}_A$, where each strategy induces a corresponding query distribution $Q_{f_A}$. Additionally, the surrogate model is evaluated on a test set drawn from a distribution $T'$. To further refine our understanding of the evaluation of the extraction process, we introduce the concept of test risk. This will help connect the attacker's observable risk on queried data to the unobservable true extraction risk.

Henceforth, under Assumption 2, we treat the extraction risk on the target distribution and the test extraction risk as equivalent. Note that an attacker has access only to the query-based risk $\tilde{R}_Q(h)$, while the true risk $\tilde{R}_T(h)$ (which reflects the ultimate success of the attack) remains unobservable. To more deeply understand the process of model extraction attacks, we need to formally define how an attacker utilizes the data obtained through queries and how they incrementally build and optimize their stolen model. The upcoming Definitions 4 5 6 will progressively construct this process.

**Definition 4 (Partition of Labeled Query Data)** *Let $\mathcal{Q} = \{x_{Q,j}, y_{Q,j}\}_{j=1}^{n_Q}$, with labels $y_{Q,j} = h_T(x_{Q,j})$. A partition of $\mathcal{Q}$ is a collection of non-empty, disjoint subsets $\{w_i\}_{i=0}^{t-1}$ such that:*

$$
\textit{1. } w_i \subseteq \mathcal{Q}, (w_i \neq \emptyset) \quad \textit{2. } w_i \cap w_j = \emptyset, \forall i \neq j \quad \textit{3. } \bigcup_{i=0}^{t-1} w_i = \mathcal{Q}. \tag{1}
$$

Although this partition is globally defined, each subset $w_i$ can be generated by a specific query strategy $f_{Ai} \in \mathcal{F}_A$, resulting in a composite strategy $f_A = \{f_{A0}, f_{A1}, \cdots, f_{A(t-1)}\}$. With these formalisms for the extraction process (see Def 5 and Def 6 in Appendix), evaluation metrics, and the iterative nature of attacks against GNNs, we can now define the problem addressed in this work.

**Problem 1** *Let $h_T$ be a fixed target GNN model trained on samples $\{(x_{T,i}, y_{T,i})\}_{i=1}^{n_T}$ drawn from the target distribution $T$. An attacker is allowed to query $h_T$ using a strategy $f_A$ to construct a labeled query dataset $\mathcal{Q}_{f_A}$ (consisting of queried graphs and their labels from $h_T$). The attacker's goal is to find an optimal strategy $f_A$ along with a corresponding partition $\{w_i\}_{i=0}^{t-1}$ and training steps $\{s_i\}_{i=0}^{t-1}$, that minimizes the extraction risk with respect to the target distribution:*

$$
\min_{\substack{w_0, w_1, \ldots, w_{t-1} \\ s_0, s_1, \ldots, s_{t-1} \\ Q_{f_A}}} \tilde{R}_T(h_t). \tag{2}
$$

## 3 THEORETICAL ANALYSIS

Solving Problem 1 explicitly by programming is a formidable challenge. The attacker must optimize over a vast search space. Furthermore, the objective function $\tilde{R}_T(h_t)$ is not directly observable by the attacker, who can only evaluate models on the query distribution $Q$. Given these complexities, we first explore general conclusions regarding the relationships between different risk measures. Subsequently, we will specialize our analysis to the graph setting. Our overarching goal is to theoretically understand the fundamental limits and critical factors influencing the performance of model extraction attacks under realistic conditions, particularly against GNNs. To this end, we begin by presenting foundational results that connect different risk measures.

**Proposition 1** $\exists \lambda_\ell, \Lambda_\ell > 0$, *s.t.*

$$\lambda_\ell |\tilde{R}_\mu(h) - R_\mu(h_T)| \leq R_\mu(h) \leq \Lambda_\ell(\tilde{R}_\mu(h) + R_\mu(h_T)), \tag{3}$$

*where the constants $\lambda_\ell, \Lambda_\ell$ depend on the loss function $\ell$, and the measure $\mu$ may be either $Q$ or $T$.*

> **NOTE.** $\lambda_\ell$ lower-bounds how much ordinary task risk must remain once a surrogate has matched the victim's outputs, whereas $\Lambda_\ell$ upper-bounds the same quantity. Both depend on the loss function (e.g., 0-1, MSE, cross-entropy) but *not* on the particular attacker.

Intuitively, Proposition 1 establishes a crucial relationship between the extracted risk $\tilde{R}_\mu(h)$ (how well the surrogate $h$ mimics the victim $h_T$) and the standard task risk $R_\mu(h)$ (how well $h$ performs on the actual task). This proposition highlights that the attacker's ability to achieve low task risk is fundamentally tied to their success in replicating the victim model's outputs, bounded by factors inherent to the chosen loss function. The absolute value in the lower bound of Proposition 1 naturally leads to two distinct cases. And it follows Theorem 1 below.

**Theorem 1** $\forall h \in \mathcal{H}$, *and for* $\mu \in \{Q, T\}$, $\exists \lambda_\ell, \Lambda_\ell > 0$ *which depend on the loss function $\ell$, s.t*

$$\begin{cases} \lambda_\ell(R_\mu(h_T) - \tilde{R}_\mu(h)) \leq R_\mu(h) \leq \Lambda_\ell(\tilde{R}_\mu(h) + R_\mu(h_T)), & R_\mu(h_T) > \tilde{R}_\mu(h). \\ R_\mu(h) = \Lambda_\ell(\tilde{R}_\mu(h) + R_\mu(h_T)), & R_\mu(h_T) \leq \tilde{R}_\mu(h). \end{cases} \tag{4}$$

We further clarify notations by using $\lambda_\ell, \Lambda_\ell$ when $\mu = Q$ and $\lambda'_\ell, \Lambda'_\ell$ when $\mu = T$. Under Assumption 1, to quantify how successful an extraction attack can be, we propose the Theorem 2.

**Theorem 2** *Given* $\lambda_\ell, \Lambda_\ell, \lambda'_\ell, \Lambda'_\ell > 0$, *then we have*

$$\tilde{R}_T(h) \geq \frac{\Lambda'_\ell}{2\lambda_\ell - \Lambda'_\ell} R_T(h_T) + \frac{1}{2\lambda_\ell - \Lambda'_\ell} \Delta_h(Q, T). \tag{5}$$

This bound depends on two main factors: (1) the inherent performance of the victim model on its own training distribution $R_T(h_T)$, which represents a baseline level of error, and (2) the generalization discrepancy $\Delta_h(Q, T)$, which captures how differently the surrogate model performs on the attacker's query distribution versus the true target distribution. Thus, we directly have a following Corollary 1.

**Corollary 1** *If an attacker can design queries such that $\Delta_h(Q, T) \to 0$, then Theorem 2 implies $\tilde{R}_T(h) \geq (\Lambda'_\ell/(2\lambda_\ell - \Lambda'_\ell))R_T(h_T)$. In this scenario, the unavoidable extraction risk is essentially a constant multiple of the target model's own risk on its native distribution.*

Here we establish a lower bound for $\tilde{R}_T(h)$. Theorem 2 indicates that the unavoidable extraction risk is bounded below by the victim's own risk and the discrepancy term; hence, the actionable goal for an attacker is to drive discrepancy toward zero. Proposition 2 further suggests an incremental route: choose partitions and intermediate models that steadily reduce risk on each partition, thereby shrinking discrepancy. These insights directly motivate TRIDENT to realize more, smaller-update steps under a fixed budget, and to prioritize hard regions where discrepancy concentrates.

**Proposition 2** *Consider the incremental extraction process defined above, for any $1 \leq k \leq t$, the surrogate model with the highest fidelity according to given condition is $h_k$ realizing*

$$\inf_{1 \leq k \leq t} \Delta_{h_k}(Q, T) = \inf_{1 \leq k \leq t} \sum_{i=0}^{t-1} \left(\frac{R_T(h_k)}{t} - R_{w_i}(h_k)\right). \tag{6}$$

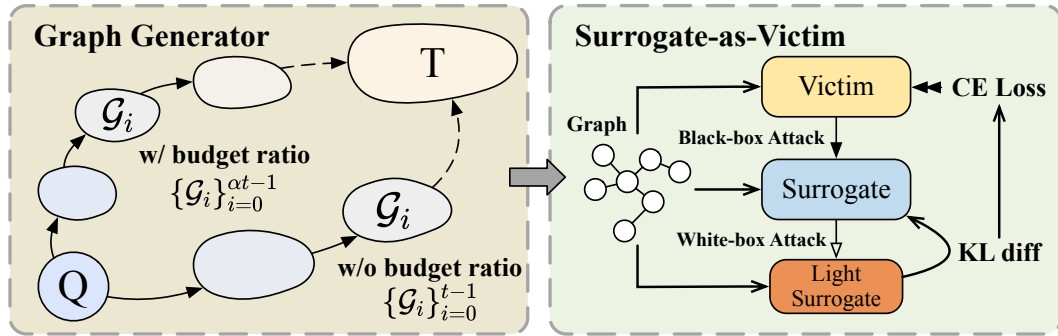

Figure 1: An illustration of proposed modules with graph generator and the alignment mechanism, which reveal that a graph generator may generate more and better graphs for the alignment.

We now discuss how these general principles can be applied in the specific context of MEAs against GNNs. Consider a graph $\mathcal{G} = (\mathcal{V}, \mathcal{E})$ to be the training data $T$. Proposition 2 suggests that during an incremental extraction attack, the attacker should aim to select query data partitions $w_i$ and train intermediate models $h_k$ in a way that minimizes the risk $R_{w_i}(h_k)$ on these partitions, while hoping this reflects a small $R_Q(h_k)$ and subsequently a small $\Delta_{h_k}(Q, T)$. Thus, attacker's first objective is to identify a set of graphs $\{\mathcal{G}_i\}_{i=0}^M$ that approximates the target graph $\mathcal{G}$ as closely as possible. In the subsequent phase (i.e., partitions $\{\mathcal{G}_i\}_{i=M+1}^{t-1}$), attackers seek to minimize $\Delta_{h_k}(Q, T)$ by reducing $R_{w_i}(h_k)$. This strategy is justified by the fact that, the attacker can only control $w_i$ and $h_k$, while $R_T(h_k)$ remains unobservable. We instantiate the target model $\mathcal{M}(\mathcal{G}; \theta)$ trained on graph $\mathcal{G}$. And we further instantiate that such surrogate model belongs to a sequence $\{\mathcal{M}'_i(\mathcal{G}_i; \theta_i)\}_{i=0}^{t-1}$, where $\mathcal{G}_i = (\mathcal{V}_i, \mathcal{E}_i)$. We note that $\mathcal{M}'_i(\mathcal{G}_i; \theta_i)$ is trained on input-output pairs $(\mathcal{G}'_i, \mathcal{M}(\mathcal{G}'_i))$, where $\mathcal{G}'_i \subset \mathcal{G}_i$. This suggests that the target model can not only supports the graph-level query, but also allows the subgraph-level query or node-level query, providing a more realistic setting.

## 4 METHODOLOGY

Our proposed TRIDENT, first applies a budget ratio to the graph generator for each step. We expect that even with less queries for each generated graph, the generator would generate higher quality graphs due to more iterations. We also propose a surrogate-as-victim module to introduce the white-box setting, thus enhancing the surrogate model by integrating KL divergence between the two surrogate models. The core theoretical motivation, as highlighted in Section 3, is to minimize the extraction risk $R_T(h)$ by effectively reducing the generalization discrepancy $\Delta_h(Q, T)$. Modules depicted above is shown in Figure 1. Specifically, we adopt the graph generator with budget ratio for all the training steps, while the alignment mechanism is implemented after a pre-heat stage. Next two subsections show how these two modules work.

### 4.1 GRAPH GENERATOR

Guided by Theorem 2, which states that minimizing generalization discrepancy significantly reduces extraction risk, our graph generator focuses on adaptively generating informative graph structures for querying. To manage the query budget $\mathcal{B}$ effectively, we introduce a budget ratio $\alpha$. Let $|\mathcal{G}_i|$ denote the number of nodes in a generated graph $\mathcal{G}_i$. Instead of querying all $|\mathcal{G}_i|$ nodes from an original set of $t$ graphs, our strategy reallocates the budget. We now generate $\alpha t$ graphs and query only $|\mathcal{G}i|/\alpha$ nodes from each, thereby aiming to keep the total number of queried nodes approximately equal to $\mathcal{B}$ (i.e., $\sum_{j=1}^{\alpha t}(|\mathcal{G}_j|/\alpha) \approx \mathcal{B}$). This approach is based on the intuition that querying a subset of nodes from a larger number of graphs, especially in early iterations, is more efficient for training as it allows for $\alpha t$ training iterations. At each iteration $i$, the graph generator $M_{\varphi_i} : \mathcal{Z} \to \mathcal{G}$ parameterized by $\varphi_i$ and taking a noise vector $z_i \sim \mathcal{N}(0, I)$, synthesizes a graph:

$$\mathcal{G}_i = (\mathcal{V}_i, A_i, X_i) \; := \; M_{\varphi_i}(z_i). \tag{7}$$

The synthesis of $\mathcal{G}_i$ involves three primary steps:

$$X_i = \psi_{\varphi_i}(z_i) \in \mathbb{R}^{n \times d}, \tag{8}$$

$$(\bar{X}_i)_{uk} = \frac{(X_i)_{uk}}{[\sum_{j=1}^{d}((X_i)_{uj}^2)]^{1/2}}, \tag{9}$$

$$(A_i)_{uv} = \mathbb{I}\left((\bar{X}_i \bar{X}_i^T)_{uv} > \tau \wedge u \neq v\right). \tag{10}$$

Here, $\psi_{\varphi_i}$ represents the feature generation network parameterized by $\varphi_i$, mapping the noise vector $z_i$ to the node feature matrix $X_i$. The adjacency matrix $A_i$ is constructed by the similarity between the normalized feature vectors of all node pairs, implemented by matrix multiplication. Once graph $\mathcal{G}_i$ is synthesized, the following procedures are executed as part of the pre-heat stage in iteration $i$:

Given current surrogate model $\mathcal{M}_i'(\mathcal{G}_i; \theta_i)$ and generated graph $\mathcal{G}_i = (\mathcal{V}_i, A_i, X_i)$, we randomly select a query node set $\mathcal{Q}_i \subset \mathcal{V}_i$ by querying $|\frac{\mathcal{G}_i}{\alpha}|$ number of nodes. The API is invoked on the induced sub-graph $\mathcal{G}_i' = \mathcal{G}_i[\mathcal{Q}_i]$ to receive hard labels $\mathcal{Y}_i = \mathcal{M}(\mathcal{G}_i')$.

During this pre-heat stage, the acquired labels $\mathcal{Y}_i$ are used to train both the surrogate model and generator. We update them by using the Cross-Entropy loss:

$$\theta_{i+1} = \theta_i - \eta_s \nabla_\theta \mathcal{L}_{\mathrm{CE}}\left(\mathcal{M}_i'(\mathcal{G}_i'; \theta_i), \mathcal{Y}_i\right). \tag{11}$$

$$\varphi_{i+1} = \varphi_i + \eta_g \nabla_\varphi \mathcal{L}_{\mathrm{CE}}\left(\mathcal{M}_i'(\mathcal{G}_i'; \theta_i), \mathcal{Y}_i\right). \tag{12}$$

Through budget ratio, we successfully expand the minimizing problem from Proposition 2 to:

$$\inf_{1 \leq k \leq t} \Delta_{h_k}(Q, T) \Rightarrow \inf_{1 \leq k \leq \alpha t} \Delta_{h_k}(Q, T). \tag{13}$$

By $M$ steps of pre-heating, we consider the generated graph is acceptably close to the target graph, and then we seek to minimize $\Delta_{h_k}(Q, T)$ in the following $\alpha t - M$ steps in alignment.

## 4.2 SURROGATE-AS-VICTIM MODULE

Our Surrogate-as-Victim Module leverages the attacker's white-box access to their own main surrogate model $\mathcal{M}_i'(\mathcal{G}_i; \theta_i)$. We propose training a light surrogate model $\overline{\mathcal{M}}_i(\mathcal{G}_i; \Theta_i)$, which shares the same backbone as the surrogate with fewer parameters. This light surrogate treats the main surrogate $\mathcal{M}_i'$ as a "pseudo-victim", while also directly learning from the true victim model's labels $\mathcal{Y}_i$. This internal interaction with light surrogate mirrors the mechanism of a white-box attack, enabling query-free training signals. For efficiency, a part of the light surrogate's weights $\Theta_i$ can be initialized from $\theta_i$, only the final layer of the light surrogate is re-initialized at each refinement step, whereas other layers inherit parameters from the surrogate. The light surrogate $\overline{\mathcal{M}}_i$ is trained using a combined loss:

$$\Theta_{i+1} = \Theta_i - \eta_l \nabla_\Theta[\xi \times \mathcal{L}_{\mathrm{KL}}(\overline{\mathcal{M}}_i(\mathcal{G}_i'; \Theta_i), \mathcal{M}_i'(\mathcal{G}_i'; \theta_i)) + \mathcal{L}_{\mathrm{CE}}\left(\mathcal{M}_i'(\mathcal{G}_i'; \theta_i), \mathcal{Y}_i\right)]. \tag{14}$$

In this formulation, $\mathcal{L}_{\mathrm{KL}}$ denotes the KL divergence, encouraging $\overline{\mathcal{M}}_i$ to mimic the output distribution of $\mathcal{M}_i'$. The $\mathcal{L}_{\mathrm{CE}}$ term trains $\overline{\mathcal{M}}_i$ using the true labels $\mathcal{Y}_i$ obtained from querying the victim model $\mathcal{M}$. The hyperparameter $\xi$ balances these two objectives. This update is query-efficient (query-free) as it reuses $\mathcal{Y}_i$, and computationally inexpensive due to $\overline{\mathcal{M}}_i$'s smaller architecture. The interaction within this setup informs the refinement of the main surrogate model $\mathcal{M}_i'$. We update $\mathcal{M}_i'$ as follows:

$$\theta_{i+1} = \theta_i - \eta_s \nabla_\theta \boldsymbol{W}_{\mathrm{KL}} \mathcal{L}_{\mathrm{CE}}\left(\mathcal{M}_i'(\mathcal{G}_i'; \theta_i), \mathcal{Y}_i\right). \tag{15}$$

The weights $\boldsymbol{W}_{\mathrm{KL}} = W_j$ are computed for each sample $j$ in the queried subgraph $\mathcal{G}_i'$. $W_j$ is derived from a temperature-scaled softmax function applied to the KL divergence between the output distributions of $\overline{\mathcal{M}}_i$ and $\mathcal{M}_i'$ for that sample: $w_{\mathrm{KL},j} = \mathrm{KL}(P_{\overline{\mathcal{M}}_{i,j}} \| P_{\mathcal{M}_{i,j}'})$. Thus, $W_j = \mathrm{softmax}(w_{\mathrm{KL},j}/T)$, normalized across samples. The KL-weighted Cross-Entropy loss, $\boldsymbol{W}_{\mathrm{KL}} \mathcal{L}_{\mathrm{CE}}$, is then defined as the mean of these individually weighted per-sample cross-entropy losses:

$$\boldsymbol{W}_{\mathrm{KL}} \mathcal{L}_{\mathrm{CE}} = \mathrm{mean}(W_j \cdot \mathcal{L}_{\mathrm{CE},j}(\mathcal{M}_i'(\mathcal{G}_{i,j}'; \theta_i), \mathcal{Y}_{i,j})). \tag{16}$$

Concurrently with refining the surrogate models, the graph generator $M_{\varphi_i}$ is also updated during this alignment stage. To foster continuous improvement in the utility of generated graphs, we employ the

same update rule for the generator as in the pre-heat stage. This continued training helps the generator produce graphs that are increasingly effective for extracting information via the improving surrogate model. The rationale for the KL-weighted loss is that $\text{KL}(P_{\overline{\mathcal{M}}_{i,j}} \parallel P_{\mathcal{M}'_{i,j}})$ quantifies the difference in predictions between the main and light surrogate models. A high KL divergence indicate samples where $\mathcal{M}'_i$ is less certain. By upweighting the CE loss for these "harder" samples, $\mathcal{M}'_i$ is guided to focus on these uncertain predictions. Conversely, samples with low KL divergence are those where $\mathcal{M}'_i$'s predictions are mimicked by $\overline{\mathcal{M}}_i$, suggesting these are either well-learned or less critical.

Drawing from Proposition 2, let $h_k$ represent the main surrogate $\mathcal{M}'_i$ and $\hat{h}_k$ the light surrogate $\overline{\mathcal{M}}_i$, where $\hat{h}_k = \Gamma(h_k)$ signifies that $\hat{h}_k$ inherits predictive properties from $h_k$ subject to its capacity constraint $\Gamma(\cdot)$. By emphasizing areas of disagreement (higher KL), we aim to reduce the risk associated with the more complex or less certain aspects of the target model, effectively focusing on $[\Delta h_k(Q,T) - \lambda \Delta \hat{h}_k(Q,T)]$, where $\Delta \hat{h}_k(Q,T)$ can be seen as the risk component captured by the simpler model and $\lambda$ a weighting factor. This refines the objective for the alignment phase $(M < k \le \alpha t)$, which compels the surrogate model to concentrate on "hard predictions."

$$\inf_{1 \le k \le \alpha t} \Delta h_k(Q,T) \quad \Rightarrow \quad \inf_{M < k \le \alpha t}[\Delta_{h_k}(Q,T) - \lambda \Delta_{\hat{h}_k}(Q,T)]. \tag{17}$$

## 5 EXPERIMENTAL EVALUATIONS

We conduct a series of experiments to evaluate the performance of TRIDENT. Specifically, we seek to address the following research questions: **RQ1:** How well can TRIDENT conduct extraction compared to existing SOTA alternatives? **RQ2:** How efficient is TRIDENT given query budgets? **RQ3:** How does each individual component contribute to the overall performance of TRIDENT?

### 5.1 EXPERIMENT SETUP

**Downstream Task and Datasets.** We adopt the widely studied node classification task as the downstream task. We perform experiments on six real-world datasets, including Cora, CiteSeer, PubMed, A-Computers, A-Photos and OGB-Arxiv. Specifically, Cora, CiteSeer, PubMed, and OGB-Arxiv are citation networks, where nodes represent research publications and edges denote the citation relationship between any two publications. A-Computers and A-Photos are two co-purchase networks from Amazon, where nodes correspond to products, and edges indicate that two products are frequently purchased together. More details can be found in Appendix C.1.

**Baselines.** To achieve fair comparison and show our state-of-the-art performance, our experiment primarily follows previous work Hu et al. (2020); Shchur et al. (2018) and aligns with the settings in the latest research. We compare TRIDENT with some baseline models as follows. Specifically, we align with the baselines provided in Zhuang et al. (2024). *(1) Real Data:* We directly use training data to train the MEA, which reveals the process of model extraction. *(2) Random Graph:* We generate random graphs without updating parameters for model extraction attack. *(3) Type III attack with cosine similarity:* Zhuang et al. (2024) We adopt the state-of-the-art model to compare the performance of our framework. Specifically, to achieve the fair comparison, we take the Type III attack as our baseline. This is because Type I attack requires a large amount of memory, and both Type I and Type II attacks each implement a surrogate models, which prevents a fair comparison with our framework. *(4) Type III attack with full parameterization:* Zhuang et al. (2024) Another implementation of Type III attack, which is reported to have a better performance than the cosine version. We note that although prior works Wu et al. (2022); Shen et al. (2022) discussed MEAs on GNNs to some degree, they require access to the training data or distribution, therefore they cannot be adopted as baselines. While other data-free methods Truong et al. (2021); Zhang et al. (2022b) provide some insights, they fail to adapt to MEAs on GNNs.

**Target Models and Surrogate Models.** To evaluate the performance of our framework, we utilize three backbone GNNs, namely GCN Kipf & Welling (2016), GAT Veličković et al. (2017), and GraphSAGE Hamilton et al. (2017). According to the model analysis in Zhuang et al. (2024), for each dataset, we select the architecture of the corresponding target model with the highest test accuracy. Additionally, we also detail defense strategies against MEAs in Appendix D.4.

**Evaluation Metrics.** We evaluate our framework by following performance metrics for MEAs Wu et al. (2022); Shen et al. (2022); Zhuang et al. (2024): *(1) Accuracy:* We evaluate the accuracy of the

Table 1: Experimental results for accuracy and fidelity of the surrogate model from attacks.

| Model | Method | Cora | | CiteSeer | | PubMed | | A-Computers | |
|---|---|---|---|---|---|---|---|---|---|
| | | Accuracy | Fidelity | Accuracy | Fidelity | Accuracy | Fidelity | Accuracy | Fidelity |
| GAT | Real Data | $26.98 \pm 6.97$ | $23.02 \pm 4.94$ | $20.88 \pm 3.32$ | $19.66 \pm 2.83$ | $40.08 \pm 2.50$ | $39.25 \pm 5.14$ | $37.94 \pm 1.60$ | $61.36 \pm 4.71$ |
| | Random Graph | $12.34 \pm 3.56$ | $10.23 \pm 2.89$ | $9.56 \pm 1.78$ | $8.12 \pm 2.44$ | $25.37 \pm 3.11$ | $22.48 \pm 4.02$ | $18.76 \pm 1.35$ | $30.45 \pm 3.67$ |
| | Attack III-A | $75.90 \pm 0.88$ | $83.75 \pm 3.00$ | $46.51 \pm 4.53$ | $56.23 \pm 5.31$ | $75.90 \pm 1.04$ | $90.04 \pm 1.70$ | $48.64 \pm 3.52$ | $76.39 \pm 1.60$ |
| | Attack III-F | $77.59 \pm 1.57$ | $88.61 \pm 2.15$ | $49.51 \pm 3.92$ | $60.43 \pm 3.41$ | $77.25 \pm 0.49$ | $87.91 \pm 1.70$ | $36.28 \pm 2.12$ | $63.36 \pm 0.37$ |
| | TRIDENT | $\mathbf{80.54 \pm 0.20}$ | $\mathbf{91.60 \pm 0.10}$ | $\mathbf{71.73 \pm 0.28}$ | $\mathbf{87.77 \pm 1.59}$ | $\mathbf{80.08 \pm 0.08}$ | $\mathbf{95.53 \pm 0.05}$ | $\mathbf{64.60 \pm 0.49}$ | $\mathbf{83.06 \pm 0.88}$ |
| GCN | Real Data | $29.89 \pm 3.58$ | $24.29 \pm 2.07$ | $21.91 \pm 1.04$ | $20.49 \pm 1.29$ | $42.73 \pm 3.79$ | $39.50 \pm 4.66$ | $37.70 \pm 0.86$ | $63.19 \pm 3.50$ |
| | Random Graph | $15.02 \pm 2.14$ | $10.57 \pm 1.83$ | $13.45 \pm 1.21$ | $12.33 \pm 1.74$ | $35.60 \pm 3.12$ | $30.22 \pm 2.55$ | $25.87 \pm 2.00$ | $50.11 \pm 4.40$ |
| | Attack III-A | $79.44 \pm 0.37$ | $89.93 \pm 1.72$ | $56.15 \pm 3.57$ | $68.29 \pm 3.82$ | $78.42 \pm 0.66$ | $89.40 \pm 0.14$ | $40.35 \pm 2.07$ | $54.12 \pm 6.52$ |
| | Attack III-F | $76.06 \pm 2.08$ | $87.14 \pm 2.42$ | $56.53 \pm 6.60$ | $72.51 \pm 3.71$ | $77.50 \pm 0.72$ | $91.03 \pm 2.84$ | $55.72 \pm 3.99$ | $82.30 \pm 4.30$ |
| | TRIDENT | $\mathbf{80.26 \pm 0.39}$ | $\mathbf{91.44 \pm 1.21}$ | $\mathbf{70.75 \pm 0.40}$ | $\mathbf{86.39 \pm 2.24}$ | $\mathbf{79.94 \pm 0.11}$ | $\mathbf{95.25 \pm 0.42}$ | $\mathbf{61.08 \pm 0.90}$ | $\mathbf{86.86 \pm 0.50}$ |
| SAGE | Real Data | $27.48 \pm 4.76$ | $23.23 \pm 5.66$ | $19.00 \pm 2.04$ | $18.62 \pm 2.79$ | $38.04 \pm 4.38$ | $35.52 \pm 6.58$ | $37.00 \pm 2.93$ | $45.11 \pm 3.65$ |
| | Random Graph | $15.87 \pm 3.21$ | $18.20 \pm 4.12$ | $17.51 \pm 2.60$ | $16.75 \pm 3.12$ | $25.77 \pm 4.89$ | $23.37 \pm 5.10$ | $7.78 \pm 1.78$ | $0.96 \pm 0.41$ |
| | Attack III-A | $73.41 \pm 4.50$ | $80.80 \pm 6.65$ | $60.17 \pm 4.21$ | $66.23 \pm 4.43$ | $79.05 \pm 0.42$ | $93.82 \pm 1.44$ | $55.78 \pm 2.14$ | $83.05 \pm 2.40$ |
| | Attack III-F | $76.84 \pm 2.48$ | $85.79 \pm 4.11$ | $55.69 \pm 4.85$ | $65.98 \pm 3.25$ | $63.06 \pm 1.86$ | $70.33 \pm 2.42$ | $47.09 \pm 3.24$ | $72.17 \pm 2.73$ |
| | TRIDENT | $\mathbf{80.00 \pm 0.40}$ | $\mathbf{91.22 \pm 0.71}$ | $\mathbf{69.20 \pm 0.65}$ | $\mathbf{86.24 \pm 0.83}$ | $\mathbf{79.53 \pm 0.07}$ | $\mathbf{94.87 \pm 0.73}$ | $\mathbf{61.42 \pm 0.91}$ | $\mathbf{86.50 \pm 0.47}$ |

surrogate model on the test dataset, which is computed based on the correct predictions measured on the ground truth. *(2) Fidelity:* We measure the consistency between the surrogate model and target model by computing fidelity. We note that fidelity essentially represents the similarity of the decision boundary between the surrogate model and target model. *(3) Query Efficiency:* We measure the fidelity of the surrogate model within the same amount of query budgets, and the number of queries that the surrogate model needs to achieve a certain fidelity.

## 5.2 EVALUATION OF EXTRACTION PERFORMANCE

To address **RQ1**, we evaluate the performance of the proposed TRIDENT by comparing both the accuracy and the fidelity of surrogate models resulting from it and multiple baselines. As explained in the experimental setup, we mainly follow the prior work Zhuang et al. (2024), which introduces real data, random graph, and Type III attack with two variants. To maintain clarity, we append the suffix "-A" and "-F" to the names of attack variants in Table 9 to represent the two different variants respectively. Here, to evaluate the best extraction performance for

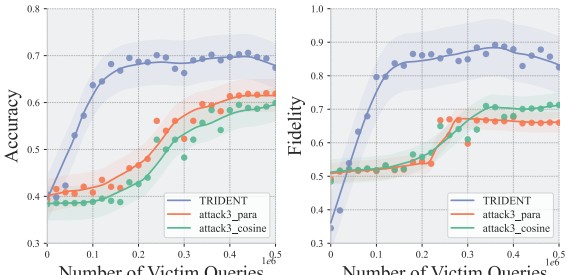

Figure 2: Left: Accuracy on MEAs over CiteSeer. Right: Fidelity on MEAs over CiteSeer.

each kind of attack, we set the query budget large enough to ensure that each kind of attack has achieved its best performance. We note that the best performance of the attack is determined by the mean of the fidelity or accuracy for given queries, where they fail to show a significant increase as the number of queries increases. Due to space limits, we present the best performance on Cora, CiteSeer, PubMed, and A-Computers, for TRIDENT and baselines targeting three backbones in Table 9. Results on the remaining datasets are provided in the Appendix. Since Zhuang et al. (2024) didn't release an official open-source code, the performance reported here might be different from that in the paper. According to Table 9, TRIDENT nearly achieves state-of-the-art performance across all the baselines on both accuracy and fidelity.

## 5.3 EVALUATION OF QUERY EFFICIENCY

To answer **RQ2**, we conduct a comprehensive evaluation from two perspectives. Firstly, for a given number of queries, we present a query-fidelity figure to show the efficiency. Secondly, for a certain fidelity, we report in Table 2 the number of queries that the surrogate model needed to achieve it. We record the mean and std value of the first 10 queries for each attack when its fidelity exceeds a threshold. We note that we present N/A here since the two methods cannot achieve the given fidelity for even infinite queries. Due to the space limitations, we only present partial results here, while the remainder is available in the Appendix. By presenting these in Figure 2 and Table 2, we derive the following key observations: (1) For a given number of queries, TRIDENT achieves not only the highest performance, but also presents a high level of efficiency. For the first few queries, TRIDENT

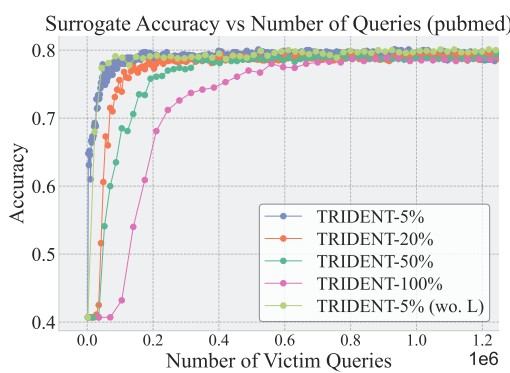 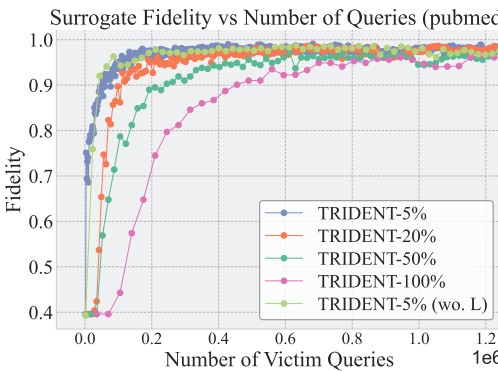

(a) Accuracy on surrogate model with ablated versions.    (b) Fidelity on surrogate model with ablated versions.

Figure 3: Comparison of accuracy and fidelity versus number of queries across the full model (5%), ablated model (100%), and other three ablated versions.

is the quickest to achieve a high performance. (2) For a certain fidelity, TRIDENT requires far fewer number of victim queries to achieve the given fidelity than other baselines. These two observations lead to the conclusion that TRIDENT is the most efficient attack among the baselines.

## 5.4 EVALUATION OF ABLATION STUDY

To address **RQ3**, we investigate the performance of the full TRIDENT framework and its ablated counterparts to assess the contribution of individual components. Specifically, we set TRIDENT with the light surrogate model and $\alpha = 20$ as our full framework. As we defined above, $\alpha = 20$ means that TRIDENT makes $\frac{1}{\alpha}$ of the queries assigned to each iteration, and we adopt these two equivalent notations in the legend of figures (e.g., TRIDENT-5%)

Table 2: Number of queries (for given fidelity) of TRIDENT and baselines over CiteSeer on GCN. The best results are highlighted in bold.

| Model | Cora ($\times 10^4$) | Citeseer ($\times 10^5$) | PubMed ($\times 10^3$) |
|---|---|---|---|
| Real Data | N/A | N/A | N/A |
| Random Graph | N/A | N/A | N/A |
| Attack III-A | $14.4 \pm 3.75$ | $3.8 \pm 0.52$ | $3.42 \pm 0.68$ |
| Attack III-F | $15.54 \pm 2.74$ | $3.74 \pm 0.16$ | $4.45 \pm 0.12$ |
| TRIDENT | $\mathbf{6.65 \pm 0.62}$ | $\mathbf{1.12 \pm 0.11}$ | $\mathbf{2.35 \pm 0.37}$ |

and the following context (e.g., $\alpha = 20$). We present the ablated framework for $\alpha = 1$, and the full framework without the light surrogate model as another ablated version to evaluate the effect of removing the alignment. To evaluate the effect of the budget ratio, we also present the full framework with $\alpha = 5$ and $\alpha = 2$ in Figure 3a and Figure 3b. The following key observations are drawn from this evaluation: (1) The incorporation of the budget ratio significantly improves both the quality and efficiency to TRIDENT. Regarding quality, we observe that a higher budget ratio always leads to better fidelity performance. Regarding efficiency, for given number of victim queries (e.g. 200000) in Figure 3a and Figure 3b, the ablated version (i.e., TRIDENT-100%) yields the lowest fidelity among all the versions. (2) The light surrogate model also contributes to the efficiency to TRIDENT, and slightly increases the best accuracy and fidelity that can be achieved. This observation agrees with our goal that the light surrogate model is designed for a cost-efficient and query-free refinement, while the budget ratio provides the primary improvement in efficiency and performance.

## 6 CONCLUSION

In this paper, we proposed TRIDENT, a query-efficient MEAs against GNNs. We further conducted a comprehensive theoretical analysis to establish a foundational understanding of the critical factors influencing MEA performance against GNNs. Extensive experiments on six real-world datasets and three GNN backbones demonstrate TRIDENT's superior performance, achieving state-of-the-art results in terms of both model fidelity and query efficiency compared to existing baselines. Furthermore, several future directions warrant further investigation. First, we expect that our theoretical explanation could provide more insights for designing MEAs targeting DNNs. Second, developing theoretically

grounded defenses specifically against such theory-aware extraction attacks like TRIDENT will be crucial for enhancing the security of MLaaS platforms.

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

## A  PROOFS

**Assumption 1** *Following empirical findings in Oliynyk et al. (2025); Zhang et al. (2025), the target model's performance imposes a lower bound on the extraction risk: $R_T(h_T) \leq \tilde{R}_T(h)$.*

**Assumption 2** *Xian et al. (2022) Assume that both the training data for the target model and the test data used for evaluating the extracted model are drawn i.i.d. from a common underlying distribution $\mathcal{D}$, i.e., $T, T' \subset \mathcal{D}$. Then, for any $h \in \mathcal{H}$, $\tilde{R}_T(h) = \tilde{R}_{T'}(h)$.*

**Definition 5 (Model Transfer Operator)** *Fix a hypothesis space $\mathcal{H}$ (GNNs) and an initial model $h_0 \in \mathcal{H}$. Given a partition $\{w_i\}_{i=0}^{t-1}$ of the query dataset $\mathcal{Q}$, and a sequence of nonnegative integers $\{s_i\}_{i=0}^{t-1}$ representing the number of training steps, define the model transfer operator by*

$$T(w_i, s_i) : \mathcal{H} \to \mathcal{H}, \quad T(w_i, s_i)(h_i) = h_{i+1}, \tag{18}$$

*where $h_{i+1}$ is obtained by applying $s_i$ optimization steps (e.g. SGD) on $h_i$ using data in $w_i$.*

**Definition 6 (Incremental Extraction)** *Given a partition $\{w_i\}_{i=0}^{t-1}$ of the query data and corresponding training steps $\{s_i\}_{i=0}^{t-1}$, the final extracted model $h_t$ is obtained by recursively applying the model transfer operator: $h_t = T(w_{t-1}, s_{t-1}) \circ \cdots \circ T(w_0, s_0)(h_0)$.*

**Assumption 3** *For classification tasks, we specify $h_T : \mathcal{X} \mapsto l_n$ to denote the target model's output probability distribution (obtained via softmax) over $n$ classes, $h : \mathcal{X} \mapsto l_n$ to denote the extracted model's output distribution, and let $y : \mathcal{X} \mapsto l_n$ represent the (possibly smoothed) true label distribution. We assume that there exist positive constants $\delta_a, \delta_b, \delta_c$ such that $\forall x \in \mathcal{X}, \forall i \in \{1, 2, \cdots, n\}$, the following holds:*

$$h_T(x)_i \geq \delta_a, \quad h(x)_i \geq \delta_b, \quad y(x)_i \geq \delta_c. \tag{19}$$

**Assumption 4** *All relevant risks are assumed to be finite, i.e.,*

$$R_Q(h), \tilde{R}_Q(h), R_T(h), \tilde{R}_T(h) < \infty. \tag{20}$$

This Assumption 4 does not require the loss function to be uniformly bounded.

**Assumption 5** *We assume the target model $h_T$ incurs a small but non-zero risk $\varepsilon = \varepsilon_{API}$ under the target distribution $T$:*

$$R_T(h_T) = \varepsilon, \tag{21}$$

Assumption 3 ensures that the probability vectors $h_T(x), h(x)$ and $y(x)$ are uniformly bounded away from zero, thereby preventing singularities (such as $\ln(0)$) in the computation of cross-entropy and KL divergence, and enabling the use of local Lipschitz properties.

**Lemma 1** *For any $h \in \mathcal{H}$, given the definitions of $Q$ and $T$ as above, both the generalization discrepancy and the extraction generalization discrepancy satisfy an anti-symmetry property:*

$$\Delta_h(Q, T) + \Delta_h(T, Q) = 0, \quad \tilde{\Delta}_h(Q, T) + \tilde{\Delta}_h(T, Q) = 0. \tag{22}$$

**Proof.** By definition 2, we apply it to our scenario twice, and then we get

$$\Delta_h(Q,T) = R_T(h) - R_Q(h), \tag{23}$$

and

$$\Delta_h(Q,T) = R_T(h) - R_Q(h). \tag{24}$$

Thus, the equation $\Delta_h(Q,T) + \Delta_h(T,Q) = 0$ holds. Similarly, by definition 3, we can derive

$$\tilde{\Delta}_h(Q,T) = \tilde{R}_T(h) - \tilde{R}_Q(h) = -\tilde{\Delta}_h(T,Q), \tag{25}$$

that is,

$$\tilde{\Delta}_h(Q,T) + \tilde{\Delta}_h(T,Q) = 0. \tag{26}$$

$\square$

**Proposition 1** $\exists \lambda_\ell, \Lambda_\ell > 0$, s.t.

$$\lambda_\ell |\tilde{R}_\mu(h) - R_\mu(h_T)| \le R_\mu(h) \le \Lambda_\ell(\tilde{R}_\mu(h) + R_\mu(h_T)), \tag{27}$$

where $\lambda_\ell, \Lambda_\ell$ depend on the loss function $\ell$ and the $\mu$ could be either $Q$ or $T$.

**Proof.** We will only prove the case when $\mu = Q$, as it is the same for $\mu = T$. Note that under the assumptions discussed above, we are allowed to write out all the risks as a summation. Specifically,

$$R_Q(h) = \sum_{i \sim Q} \ell(h(x_i), y_i) p_i, \tag{28}$$

$$\tilde{R}_Q(h) = \sum_{i \sim Q} \ell(h(x_i), h_T(x_i)) p_i, \tag{29}$$

and

$$R_Q(h_T) = \sum_{i \sim Q} \ell(h_T(x_i), y_i) p_i. \tag{30}$$

Thus, we could further write the proposition as

$$\lambda_\ell |\ell(h(x_i), h_T(x_i)) - \ell(h_T(x_i), y_i)| \le \ell(h(x_i), y_i) \le \Lambda_\ell(\ell(h(x_i), h_T(x_i)) + \ell(h_T(x_i), y_i)) \tag{31}$$

For simplicity, we write $a$ as $h(x_i)$, $b$ as $y_i$, and $c$ as $h_T(x_i)$, and we are going to prove

$$\lambda_\ell |\ell(a,c) - \ell(c,b)| \le \ell(a,b) \le \Lambda_\ell(\ell(a,c) + \ell(c,b)) \tag{32}$$

Since this inequality primarily depends on the loss function, we only verify some common loss functions here.

1. 0-1 loss function: This is the simplest case as the 0-1 loss induces a discrete topology. Indeed we only need to take $\lambda_\ell = \Lambda_\ell = 1$.

2. L1 loss function: Similarly, L1 loss also induces a topology, and this is the most well-known triangle inequality when $\lambda_\ell = \Lambda_\ell = 1$.

3. MSE loss: Before proving the inequality, we have some observations that $a$, $b$, and $c$ belong to some bounded intervals $[m, M]$, $m > 0$, and $\forall i, \exists \delta > 0, s.t. |h(x_i) - h_T(x_i)| \ge \delta$. Empirically, we also have $|h(x_i) - y_i| \ge |h(x_i) - h_T(x_i)|$, then by taking $\lambda_\ell = \frac{\delta}{2M}, \Lambda_\ell = 2$, the inequality holds. We present the details as follows: For the left-hand side, we first rewrite the inequality by

$$(a - b)^2 \ge \frac{\delta}{2M} |a - 2c + b| |a - b|. \tag{33}$$

It is trivial for $a = b$, thus we focus on the case when $a \ne b$, then we further get

$$|a - b| \ge \frac{\delta}{2M} |a - 2c + b|. \tag{34}$$

By observation, we have

$$|a - 2c + b| = |(a + b) - 2c| \le 2M, \tag{35}$$

then

$$\frac{\delta}{2M}|a - 2c + b| \le \delta. \tag{36}$$

Also,

$$|a - b| \ge \delta, \tag{37}$$

we finally prove that

$$|a - b| \ge \frac{\delta}{2M}|a - 2c + b|. \tag{38}$$

For the right-hand side, we notice that

$$a - b = (a - c) + (c - b). \tag{39}$$

It follows that

$$(a - b)^2 = [(a - c) + (c - b)]^2. \tag{40}$$

By Cauchy-Schwartz inequality, i.e.

$$\forall x, y \in \mathbb{R}, \quad (x + y)^2 \le 2(x^2 + y^2), \tag{41}$$

taking $x = a - c, y = c - b$, we get

$$(a - b)^2 = [(a - c) + (c - b)]^2 \le 2[(a - c)^2 + (c - b)^2], \tag{42}$$

which proves the right-hand side when $\Lambda_\ell = 2$.

4. Cross-entropy loss: We note that in the following context, we use $a, b, c$ to represent the probability distribution, then by Assumption 3, $\forall i, a_i \ge \delta_a, b_i \ge \delta_b, c_i \ge \delta_c$. Since the cross-entropy loss is given by $\ell(p, q) = -\sum_{i=1}^n p_i \ln q_i$, we notice that there is a basic equation that

$$\ell(p, q) = H(p) + D_{KL}(p||q), \tag{43}$$

where $H(p)$ is the entropy of $p$, and $D_{KL}(p||q)$ is the KL-divergence between $p$ and $q$. Therefore, we could rewrite the inequality as

$$H(a) + D_{KL}(a||b) \le \Lambda_\ell (H(a) + D_{KL}(a||c) + H(b) + D_{KL}(b||c)). \tag{44}$$

We first connect $D_{KL}(a||b)$ with $D_{KL}(a||c)$ by

$$D_{KL}(a||b) = \sum_{i=1}^n a_i \ln \frac{a_i}{b_i} = \sum_{i=1}^n a_i \ln \frac{a_i c_i}{b_i c_i} = D_{KL}(a||c) + \sum_{i=1}^n a_i \ln \frac{c_i}{b_i}. \tag{45}$$

Let $\delta_{b,c} = \min\{\delta_b, \delta_c\}$, we notice that $\ln(\cdot)$ is Lipschitz-continuous at $[\delta_{b,c}, 1]$, we then have

$$|\ln \frac{c_i}{b_i}| \le \frac{|c_i - b_i|}{\delta_{b,c}}. \tag{46}$$

Thus, we derive the upper bound pf the difference between $D_{KL}(a||b)$ and $D_{KL}(a||c)$

$$|D_{KL}(a||b) - D_{KL}(a||c)| = |\sum_{i=1}^n a_i \ln \frac{c_i}{b_i}| \le \frac{1}{\delta_{b,c}} \sum_{i=1}^n |c_i - b_i|. \tag{47}$$

According to Pinsker inequality,

$$\sum_{i=1}^n |c_i - b_i| \le 2\sqrt{\frac{1}{2} D_{KL}(b||c)} = \sqrt{2 D_{KL}(b||c)}. \tag{48}$$

Then,

$$|\sum_{i=1}^n a_i \ln \frac{c_i}{b_i}| \le \frac{\sqrt{2 D_{KL}(b||c)}}{\delta_{b,c}}. \tag{49}$$

We consider a small permutation $\epsilon > 0$, such that

$$\sqrt{D_{KL}(b||c)} \le \epsilon + \frac{D_{KL}(b||c)}{4\epsilon}. \tag{50}$$

Let $C_1 = \frac{\sqrt{2}\epsilon}{\delta_{b,c}}, C_2 = \frac{\sqrt{2}}{4\delta_{b,c}\epsilon}$, we then get

$$|\sum_{i=1}^{n} a_i \ln \frac{c_i}{b_i}| \leq C_1 + C_2 D_{KL}(b||c). \tag{51}$$

We denote the infimum of the entropy for each distribution $p$ (where $p$ could be $a, b, c$) by

$$H_{\min,p} = -[(1 - (n-1)\delta_p) \ln((1 - (n-1)\delta_p)) + (n-1)\delta_p \ln \delta_p]. \tag{52}$$

It follows that

$$H(p) \geq H_{\min,p} \quad H(p) \leq \ln n. \tag{53}$$

Let the permutation $\epsilon = \frac{\sqrt{H_{\min}^{b,c}}}{2}$, where $H_{\min}^{b,c} = \min\{H_{\min,b}, H_{\min,c}\}$, we could rewrite $C_1, C_2$ by

$$C_1 = \frac{\sqrt{2H_{\min}^{b,c}}}{2\delta_{b,c}}, \quad C_2 = \frac{\sqrt{2}}{2\delta_{b,c}\sqrt{H_{\min}^{b,c}}}. \tag{54}$$

Thus, to determine the value of $\Lambda_\ell$, we just need to check when would the following inequality hold

$$H(a) + D_{KL}(a||c) + C_1 + C_2 D_{KL}(b||c) \leq \Lambda_\ell(H(a) + D_{KL}(a||c) + H(b) + D_{KL}(b||c)). \tag{55}$$

Here we choose to realize this by parts. Specifically, we need to find such $\Lambda_\ell$, which satisfies

$$C_1 \leq \Lambda_\ell[H(a) + H(b)], \tag{56}$$

$$C_2 D_{KL}(b||c) \leq (\Lambda_\ell - 1)D_{KL}(b||c). \tag{57}$$

Furthermore, $\Lambda_\ell$ itself requires that

$$\Lambda_\ell \geq \frac{\ln n}{H_{\min,a} + H_{\min,b}}. \tag{58}$$

Thus, take

$$\Lambda_\ell = \max\{\frac{\ln n}{H_{\min,a} + H_{\min,b}}, 1 + \frac{\sqrt{2}}{2\delta_{b,c}\sqrt{H_{\min}^{b,c}}}, \frac{\sqrt{2H_{\min}^{b,c}}}{2\delta_{b,c}(H_{\min,a} + H_{\min,b})}\}, \tag{59}$$

we prove that

$$\ell(a, b) \leq \Lambda_\ell(\ell(a, c) + \ell(c, b)). \tag{60}$$

Then consider the left-hand side, similarly, we rewrite the inequality as

$$|\ell(a, c) - \ell(c, b)| \leq |H(a) - H(c)| + |D_{KL}(a||c) - D_{KL}(c||b)|. \tag{61}$$

And we can derive two inequalities by following a similar procedure shown above

$$|H(a) - H(c)| \leq L_1\|a - c\|_1, \quad |D_{KL}(a||c) - D_{KL}(c||b)| \leq L_2(\|a - c\|_1 + \|c - b\|_1). \tag{62}$$

Thus we obtain

$$|\ell(a, c) - \ell(c, b)| \leq (L_1 + L_2)\|a - c\|_1 + L_2\|c - b\|_1. \tag{63}$$

According to Pinsker's inequality, we further write the inequality as

$$|\ell(a, c) - \ell(c, b)| \leq (L_1 + L_2)\sqrt{2D_{KL}(a||c)} + L_2\sqrt{2D_{KL}(c||b)}. \tag{64}$$

Recall that

$$\ell(a, b) = H(a) + D_{KL}(a||b). \tag{65}$$

Since $H(a)$ is bounded below by

$$H_{\min,a} = -[(1 - (n-1)\delta_a) \ln((1 - (n-1)\delta_a)) + (n-1)\delta_a \ln \delta_a], \tag{66}$$

we have

$$\ell(a, b) \geq H(a) \geq H_{\min,a}. \tag{67}$$

Thus, if we can choose a constant $\lambda_\ell$ satisfying

$$\lambda_\ell[(L_1 + L_2)\sqrt{2D_{KL}(a||c)} + L_2\sqrt{2D_{KL}(c||b)}] \leq H_{\min,a}, \tag{68}$$

then it follows that

$$\lambda_\ell|\ell(a,c) - \ell(c,b)| \leq \ell(a,b). \tag{69}$$

Now, note that under our assumptions the KL divergences are themselves bounded above, For simplicity, denote

$$K := \ln(\frac{1}{\delta_{b,c}}), \tag{70}$$

then we have

$$\sqrt{D_{KL}(a||c)} \leq \sqrt{K}, \quad \sqrt{D_{KL}(c||b)} \leq \sqrt{K}. \tag{71}$$

Thus,

$$|\ell(a,c) - \ell(c,b)| \leq (L_1 + 2L_2)\sqrt{2K}. \tag{72}$$

A valid choice is then to take

$$\lambda_\ell = \frac{H_{\min,a}}{(L_1 + 2L_2)\sqrt{2K}}. \tag{73}$$

Recalling that the Lipschitz constants $L_1, L_2$ can be chosen proportional to $\frac{1}{\delta_{b,c}}$, we could further specify

$$\lambda_\ell = \frac{H_{\min,a}\delta_{b,c}}{2\sqrt{\ln(\frac{1}{\delta_{b,c}})}}. \tag{74}$$

Finally, we prove that the left-hand side also holds.

$\square$

**Theorem 1** *Along with proposition 1, more specifically, $\forall h \in \mathcal{H}$, given $\mu$ to be either $Q$ or $T$, $\exists \lambda_\ell, \Lambda_\ell > 0$, s.t*

$$\begin{cases} \lambda_\ell(R_\mu(h_T) - \tilde{R}_\mu(h)) \leq R_\mu(h) \leq \Lambda_\ell(\tilde{R}_\mu(h) + R_\mu(h_T)), & R_\mu(h_T) > \tilde{R}_\mu(h). \\ R_\mu(h) = \Lambda_\ell(\tilde{R}_\mu(h) + R_\mu(h_T)), & R_\mu(h_T) \leq \tilde{R}_\mu(h), \end{cases} \tag{75}$$

*where $\lambda_\ell, \Lambda_\ell$ depend on the loss function $\ell$.*

**Proof.** Similarly, WLOG, we only consider the case when $\mu = T$. To deal with the absolute value in proposition 1, we firstly try to unify the coefficients with $0 < \varepsilon < 1$ by finding a proper value $w$, s.t.

$$\lambda_\ell'|\tilde{R}_T(h) + \varepsilon R_T(h_T) - (\varepsilon + 1)R_T(h_T)| \leq \lambda_\ell'|\varepsilon\tilde{R}_T(h) + \varepsilon R_T(h_T) - wR_T(h_T)|, \tag{76}$$

which requires that

$$w \leq 1 + \varepsilon + (\varepsilon - 1)\frac{\tilde{R}_T(h)}{R_T(h_T)}. \tag{77}$$

According to proposition 1, we have

$$\frac{\tilde{R}_T(h)}{R_T(h_T)} \geq \frac{R_T(h)}{\Lambda_\ell' R_T(h_T)} - 1, \tag{78}$$

one valid value of $w$ could be

$$w = 2 + \frac{\varepsilon - 1}{\Lambda_\ell'}\frac{R_T(h)}{R_T(h_T)}. \tag{79}$$

Substitute it into the inequality above, we get

$$\lambda_\ell'|R_T(h_T) - \tilde{R}_T(h)| \leq \lambda_\ell'|\varepsilon\tilde{R}_T(h) + (\varepsilon - 2)R_T(h_T) - \frac{\varepsilon - 1}{\Lambda_\ell'}R_T(h)|. \tag{80}$$

Then there are four cases when canceling the absolute value. For the first case, we consider $\varepsilon\tilde{R}_T(h) + (\varepsilon - 2)R_T(h_T) - \frac{\varepsilon-1}{\Lambda_\ell'}R_T(h) \geq 0$, which could be further rewritten as

$$\Lambda_\ell'(\tilde{R}_T(h) + R_T(h_T) + \frac{1}{\varepsilon - 1}(\tilde{R}_T(h) - R_T(h_T))) \leq R_T(h). \tag{81}$$

From 1, the equation holds if and only if $\tilde{R}_T(h) - R_T(h_T) \geq 0$. Thus, the inequality 80 is reduced to

$$\tilde{R}_T(h) - R_T(h_T) \leq \varepsilon \tilde{R}_T(h) + (\varepsilon - 2)R_T(h_T) - \frac{\varepsilon - 1}{\Lambda'_\ell}R_T(h), \tag{82}$$

which reveals that

$$R_T(h) \geq \Lambda'_\ell(\tilde{R}_T(h) + R_T(h_T)). \tag{83}$$

We get an interesting conclusion that

$$R_T(h) = \Lambda'_\ell(\tilde{R}_T(h) + R_T(h_T)), \tag{84}$$

when $\tilde{R}_T(h) \geq R_T(h_T)$. Similarly, we consider the case when $\varepsilon \tilde{R}_T(h) + (\varepsilon - 2)R_T(h_T) - \frac{\varepsilon - 1}{\Lambda'_\ell}R_T(h) < 0$, which could be rewritten as

$$\Lambda'_\ell(\tilde{R}_T(h) + R_T(h_T) + \frac{1}{\varepsilon - 1}(\tilde{R}_T(h) - R_T(h_T))) \geq R_T(h). \tag{85}$$

This requires that $\tilde{R}_T(h) - R_T(h_T) \leq 0$. It follows that

$$(\varepsilon - 1)R_T(h_T) + (\varepsilon - 1)\tilde{R}_T(h) \leq \frac{\varepsilon - 1}{\Lambda'_\ell}R_T(h), \tag{86}$$

i.e.

$$\Lambda'_\ell(R_T(h_T) + \tilde{R}_T(h)) \geq R_T(h). \tag{87}$$

Finally, we prove that

$$\begin{cases} \lambda_\ell(R_\mu(h_T) - \tilde{R}_\mu(h)) \leq R_\mu(h) \leq \Lambda_\ell(\tilde{R}_\mu(h) + R_\mu(h_T)), & R_\mu(h_T) > \tilde{R}_\mu(h). \\ R_\mu(h) = \Lambda_\ell(\tilde{R}_\mu(h) + R_\mu(h_T)), & R_\mu(h_T) \leq \tilde{R}_\mu(h). \end{cases} \tag{88}$$

$\square$

**Lemma 3** $\exists \lambda_\ell, \Lambda_\ell, \lambda'_\ell, \Lambda'_\ell > 0$, *s.t.*

$$\begin{cases} \lambda_\ell(R_Q(h_T) - \tilde{R}_Q(h)) \leq R_Q(h) \leq \Lambda_\ell(\tilde{R}_Q(h) + R_Q(h_T)), & R_Q(h_T) > \tilde{R}_Q(h). \\ R_T(h) = \Lambda'_\ell(\tilde{R}_T(h) + R_T(h_T)), & R_T(h_T) \leq \tilde{R}_T(h). \end{cases} \tag{89}$$

**Lemma 4** $\exists \Lambda_\ell, \Lambda'_\ell > 0$, *s.t.*

$$\begin{cases} R_Q(h) = \Lambda_\ell(\tilde{R}_Q(h) + R_Q(h_T)), & R_Q(h_T) \leq \tilde{R}_Q(h). \\ R_T(h) = \Lambda'_\ell(\tilde{R}_T(h) + R_T(h_T)), & R_T(h_T) \leq \tilde{R}_T(h). \end{cases} \tag{90}$$

**Observation 5** *In lemma 3, we have*

$$\frac{\Lambda'_\ell}{2} \leq \lambda_\ell \leq \Lambda_\ell. \tag{91}$$

One can easily verify these equations using theorems proved above.

**Lemma 5** *Given $\lambda_\ell, \Lambda_\ell, \lambda'_\ell, \Lambda'_\ell > 0$, we have two relationships between $\tilde{R}_Q(h)$ and $\tilde{R}_T(h)$ when $R_Q(h_T) > \tilde{R}_Q(h)$ and $R_Q(h_T) \leq \tilde{R}_Q(h)$ respectively,*

$$\begin{cases} \tilde{R}_Q(h) + \frac{1}{2}(\frac{1}{\Lambda_\ell} - \frac{1}{\lambda_\ell})(R_T(h) - R_Q(h)) \geq \frac{1}{2}(\frac{\Lambda'_\ell}{\Lambda_\ell} - \frac{\Lambda'_\ell}{\lambda_\ell})(\tilde{R}_T(h) + R_T(h_T)) + \tilde{R}_T(h), \\ \tilde{R}_Q(h) = \frac{\Lambda'_\ell}{\Lambda_\ell}\tilde{R}_T(h) + \frac{\Lambda'_\ell}{\Lambda_\ell}R_T(h_T) - R_Q(h_T) + \frac{1}{\Lambda_\ell}(R_Q(h) - R_T(h)). \end{cases} \tag{92}$$

**Proof.** We first prove the equation

$$\tilde{R}_Q(h) + \frac{1}{2}(\frac{1}{\Lambda_\ell} - \frac{1}{\lambda_\ell})(R_T(h) - R_Q(h)) \geq \frac{1}{2}(\frac{\Lambda'_\ell}{\Lambda_\ell} - \frac{\Lambda'_\ell}{\lambda_\ell})(\tilde{R}_T(h) + R_T(h_T)) + \tilde{R}_T(h). \tag{93}$$

From Lemma 3, since $R_Q(h) = R_T(h) + \Delta_h(T, Q)$, we then have

$$\lambda_\ell(R_Q(h_T) - \tilde{R}_Q(h)) \leq \Lambda'_\ell(\tilde{R}_T(h) + R_T(h_T)) + \Delta_h(T, Q) \leq \Lambda_\ell(\tilde{R}_Q(h) + R_Q(h_T)). \tag{94}$$

For the left-hand side, we finally reduce it to

$$\tilde{\Delta}_h(T,Q) \geq \Delta_{h_T}(T,Q) - (\frac{\Lambda'_\ell}{\lambda_\ell} + 1)\tilde{R}_T(h) - (\frac{\Lambda'_\ell}{\lambda_\ell} - 1)R_T(h_T) - \frac{\Delta_h(T,Q)}{\lambda_\ell}. \quad (95)$$

For the right-hand side, we can also reduce it to

$$\tilde{\Delta}_h(T,Q) \geq -\Delta_{h_T}(T,Q) + (\frac{\Lambda'_\ell}{\Lambda_\ell} - 1)(\tilde{R}_T(h) + R_T(h_T)) + \frac{\Delta_h(T,Q)}{\Lambda_\ell}. \quad (96)$$

According to equation 95 96, we directly have

$$2\tilde{\Delta}_h(T,Q) \geq (\frac{\Lambda'_\ell}{\Lambda_\ell} - \frac{\Lambda'_\ell}{\lambda_\ell})(\tilde{R}_T(h) + R_T(h_T)) + (\frac{1}{\Lambda_\ell} - \frac{1}{\lambda_\ell})\Delta_h(T,Q), \quad (97)$$

which essentially is the equation aimed to prove. Then consider the second equation

$$\tilde{R}_Q(h) = \frac{\Lambda'_\ell}{\Lambda_\ell}\tilde{R}_T(h) + \frac{\Lambda'_\ell}{\Lambda_\ell}R_T(h_T) - R_Q(h_T) + \frac{1}{\Lambda_\ell}(R_Q(h) - R_T(h)). \quad (98)$$

Similarly, according to Lemma 4, we have

$$\tilde{\Delta}_h(T,Q) = -\Delta_{h_T}(T,Q) + (\frac{\Lambda'_\ell}{\Lambda_\ell} - 1)(\tilde{R}_T(h) + R_T(h_T)) + \frac{\Delta_h(T,Q)}{\Lambda_\ell}, \quad (99)$$

then we finish the proof. $\qquad\square$

**Theorem 2** *Given $\lambda_\ell, \Lambda_\ell, \lambda'_\ell, \Lambda'_\ell > 0$, then we have*

$$\tilde{R}_T(h) \geq \frac{\Lambda'_\ell}{2\lambda_\ell - \Lambda'_\ell}R_T(h_T) + \frac{1}{2\lambda_\ell - \Lambda'_\ell}\Delta_h(Q,T). \quad (100)$$

**Proof.** According to Lemma 5, we first relax the second equality constraint to an inequality by applying $R_Q(h_T) \leq \tilde{R}_Q(h)$. It follows that

$$\tilde{R}_Q(h) \geq \frac{1}{2}\frac{\Lambda'_\ell}{\Lambda_\ell}(\tilde{R}_T(h) + R_T(h_T)) + \frac{1}{2\Lambda_\ell}(R_Q(h) - R_T(h)). \quad (101)$$

Also, the first equation in the proposition reveals that

$$\tilde{R}_Q(h) - \tilde{R}_T(h) \geq \frac{1}{2}(\frac{\Lambda'_\ell}{\Lambda_\ell} - \frac{\Lambda'_\ell}{\lambda_\ell})(\tilde{R}_T(h) + R_T(h_T)) + \frac{1}{2}(\frac{1}{\Lambda_\ell} - \frac{1}{\lambda_\ell})(R_Q(h) - R_T(h)). \quad (102)$$

For simplicity, we denote

$$A_1 = \frac{1}{2}\frac{\Lambda'_\ell}{\Lambda_\ell}(\tilde{R}_T(h) + R_T(h_T)), B_1 = \frac{1}{2\Lambda_\ell}(R_Q(h) - R_T(h)), \quad (103)$$

then we have

$$\tilde{R}_Q(h) \geq A_1 + B_1. \quad (104)$$

Similarly, we denote

$$A_2 = \frac{1}{2}(\frac{\Lambda'_\ell}{\Lambda_\ell} - \frac{\Lambda'_\ell}{\lambda_\ell})(\tilde{R}_T(h) + R_T(h_T)), B_2 = \frac{1}{2}(\frac{1}{\Lambda_\ell} - \frac{1}{\lambda_\ell})(R_Q(h) - R_T(h)), \quad (105)$$

then we have

$$\tilde{R}_Q(h) - \tilde{R}_T(h) \geq A_2 + B_2. \quad (106)$$

Then it follows that

$$\tilde{R}_T(h) \geq (A_1 - A_2) + (B_1 - B_2), \quad (107)$$

which is equivalent to

$$(1 - \frac{\Lambda'_\ell}{2\lambda_\ell})\tilde{R}_T(h) \geq \frac{\Lambda'_\ell}{2\lambda_\ell}R_T(h_T) + \frac{1}{2\lambda_\ell}(R_Q(h) - R_T(h)). \quad (108)$$

By observation, we further write this as

$$\tilde{R}_T(h) \geq \frac{\Lambda'_\ell}{2\lambda_\ell - \Lambda'_\ell}R_T(h_T) + \frac{1}{2\lambda_\ell - \Lambda'_\ell}\Delta_h(Q,T). \quad (109)$$

$\qquad\square$

Table 3: Key symbols used throughout Sections 2–3..

| Symbol | Meaning |
|---|---|
| $T,\ Q$ | Victim's hidden training distribution and attacker's query distribution |
| $h_T$ | Victim (target) model |
| $h$ | Generic surrogate / hypothesis |
| $\tilde{R}_\mu(h)$ | Extraction risk on distribution $\mu$ |
| $\Delta_h(\mu, \mu')$ | Generalization discrepancy between $\mu'$ and $\mu$ |
| $w_i,\ s_i$ | $i$-th query subset and its training step budget (Def. 5) |

Table 4: Key symbols used throughout Section 4

| Symbol | Meaning |
|---|---|
| $(\cdot)_{uk}/(\cdot)_{uv}$ | Element of a matrix at row $u$, column $k$ or $v$ |
| $n$ | Number of nodes |
| $d$ | Dimension of node features |
| $\mathcal{G}_i$ | Generated graph at iteration $i$ |
| $\mathcal{B}$ | Fixed total query budget |
| $t$ | Total number of iterations, implied by $\sum_{i=1}^t$ |
| $\alpha$ | Budget ratio |
| $M_{\varphi_i}$ | Graph Generator model at iteration $i$ |
| $\mathcal{V}_i$ | Set of nodes/vertices in graph $\mathcal{G}_i$ |
| $A_i$ | Adjacency matrix of graph $\mathcal{G}_i$ |
| $X_i$ | Node feature matrix of graph $\mathcal{G}_i$ |
| $\psi_{\varphi_i}$ | Feature generation network parameterized by $\varphi_i$ |
| $\mathcal{M}(\cdot)$ | Target (victim) model |
| $\mathcal{M}_i'(\cdot; \theta_i)$ | Surrogate model at iteration $i$ with parameters $\theta_i$ |
| $\overline{\mathcal{M}}_i(\cdot; \Theta_i)$ | Light surrogate model at iteration $i$ with parameters $\Theta_i$ |
| $P_{\overline{\mathcal{M}}_{i,j}}$ | Predicted probability distribution by the light surrogate for node $j$ at iteration $i$ |
| $P_{\mathcal{M}'_{i,j}}$ | Predicted probability distribution by the surrogate model for node $j$ at iteration $i$ |

## B  NOTATION TABLE

## C  REPRODUCIBILITY

This appendix details the datasets, experimental configurations, and implementation specifics required to **faithfully replicate** every result reported in the main paper. All source code, checkpoint files, and YAML configs are publicly released at `https://anonymous.4open.science/r/TRIDENT-gnn-C72A/`.

### C.1  DATASETS

We evaluate on six *publicly licensed* benchmarks for node classification; key statistics are summarised in Table 5.

### C.2  EXPERIMENTAL SETTINGS

**Hardware.**   All experiments ran on **Ubuntu 22.04 LTS**, four NVIDIA GeForce RTX 4090 GPUs (24 GiB each), and an Intel Xeon Platinum 8468 CPU (32 cores, 2.1 GHz).

**Software Environment.**   Python 3.10, PyTorch 2.2.1 with PyTorch Geometric 2.5, CUDA 12.4, cuDNN 9.0, NumPy 1.26, and SciPy 1.13.

Table 5: Statistics of benchmark graph datasets used in our study.

|  | #Nodes | #Edges | #Attributes | #Classes |
|---|---|---|---|---|
| Cora | 2,708 | 5,069 | 1,433 | 7 |
| CiteSeer | 3,312 | 4,732 | 3,703 | 6 |
| PubMed | 19,717 | 44,324 | 500 | 3 |
| A-Computers | 13,381 | 245,778 | 767 | 10 |
| A-Photos | 7,650 | 238,162 | 745 | 8 |
| OGB-Arxiv | 169,343 | 1,166,243 | 128 | 40 |

## C.3  IMPLEMENTATION DETAILS

TRIDENT is implemented in PyTorch Paszke et al. (2017) and optimized using the **Adam** Kingma & Ba (2014) optimizer ($\beta_1 = 0.9$, $\beta_2 = 0.999$, $\epsilon = 1 \times 10^{-8}$). The model is trained for 200 epochs with five different random seeds, and we report the mean and standard deviation of the results across those runs.

## C.4  BASELINES

We benchmark our detector against four state-of-the-art model extraction attacks, categorized according to the STEALGNN taxonomy:

**Real Data.** An upper-bound attack that trains the surrogate model on the victim's original training dataset.

**Random Graph.** A naïve baseline which issues queries on randomly generated graphs without iterative refinement.

**Type III–Cosine.** The standard STEALGNN attack employing a cosine-similarity–based surrogate graph generator.

**Type III–FullParam.** A more expressive variant featuring a fully parameterized surrogate graph generator.

For clarity, we append the suffix "-A" (Cosine) or "-F" (FullParam) to each baseline's name in our result tables. All baselines undergo an identical hyperparameter search (80 configurations) and are early-stopped based on validation fidelity to guarantee a fair and rigorous comparison. We adopt Type III as our primary competitor because it is the most recent SOTA data-free GNN extraction method under comparable threat models. Adapted data-free MEAs. Across node-level tasks, TRIDENT requires substantially fewer queries to hit the same fidelity, remaining SOTA in efficiency.

## C.5  HYPER-PARAMETER SETTINGS AND QUERY BUDGETS

We sweep the following hyper-parameters (best chosen on validation fidelity): **learning rate** $\in \{1 \times 10^{-3}, 5 \times 10^{-4}, 1 \times 10^{-4}\}$, **hidden dimension** $\in \{64, 128, 256\}$, **entropy coefficient** $\in \{0, 1 \times 10^{-4}, 5 \times 10^{-4}\}$; **budget ratio** $\alpha \in \{1, 2, 5, 20\}$ where $\alpha = 20$ corresponds to 5 % queries per iteration . Dataset-specific total query budgets follow Table 2 of the main paper: $B = 10^4$ (Cora), $10^5$ (CiteSeer), $10^3$ (PubMed); efficiency curves extend to $10^6$ queries where relevant.

Besides, we include a parameter sensitivity study in Table 6. Since prior work Zhuang et al. (2024) has already provided detailed analyses regarding the surrogate model size and architecture choices, here we focus on the key hyperparameter unique to TRIDENT, that is, the weight of the KL loss in the alignment module. Specifically, we sweep the KL weight in the range $[0.3, 0.7]$ to fully evaluate its effect. The selected range mainly covers the region where the KL alignment term contributes comparably to the CE loss, while avoiding extreme regimes where either term dominates the optimization dynamics. The results show that a weight of $0.5$ yields the best overall performance, and thus it is adopted as the default setting in the main paper. Both decreasing and increasing the weight lead to performance degradation. Specifically, on Cora and PubMed, larger KL weights result in noticeably worse performance than smaller ones. This phenomenon suggests that excessively emphasizing the KL divergence forces TRIDENT to over-focus on "hard" samples, while inadvertently weakening its ability to maintain stable performance on easier samples. Overall, TRIDENT exhibits reasonably

Table 6: Parameter test on the weight of KL loss on GCN backbone.

| KL weight | Cora | | CiteSeer | | PubMed | | A-Computers | |
|---|---|---|---|---|---|---|---|---|
| | Acc | Fid | Acc | Fid | Acc | Fid | Acc | Fid |
| 0.30 | 63.15 | 75.20 | 62.11 | 77.74 | 71.21 | 85.94 | 48.79 | 75.03 |
| 0.35 | 69.53 | 80.01 | 63.41 | 79.84 | 72.51 | 88.75 | 50.92 | 76.87 |
| 0.40 | 72.68 | 83.26 | 67.83 | 81.71 | 76.02 | 91.86 | 54.95 | 80.12 |
| 0.45 | 77.59 | 87.94 | 69.10 | 84.37 | 77.82 | 93.41 | 57.29 | 82.61 |
| 0.50 | 80.26 | 91.44 | 70.75 | 86.39 | 79.94 | 95.25 | 61.08 | 86.86 |
| 0.55 | 76.87 | 87.20 | 68.52 | 83.97 | 74.32 | 91.86 | 58.12 | 82.89 |
| 0.60 | 70.49 | 80.97 | 67.87 | 82.85 | 71.27 | 89.03 | 57.06 | 80.71 |
| 0.65 | 64.28 | 74.78 | 65.28 | 81.64 | 70.74 | 87.52 | 53.77 | 77.24 |
| 0.70 | 56.14 | 68.88 | 63.27 | 79.92 | 67.48 | 84.28 | 52.11 | 76.90 |

stable performance across the tested weight range, and on datasets such as CiteSeer, PubMed, and A-Computers, it continues to deliver competitive or superior results compared with baseline methods, demonstrating robustness to variations in this hyperparameter.

Experiments for **RQ1/RQ2** run on the RTX 4090 cluster; **RQ3** and defense evaluations use a single RTX 4080 Ti. The exact conda environment is exported as `environment.yml` in the repository, listing every dependency and version for full reproducibility.

# D  APPENDIX

## D.1  RELATED WORKS

**MEAs in Graph Domains.** Prior work Wu et al. (2022); DeFazio & Ramesh (2019) has primarily explored MEA targeting GNNs within **transductive** learning settings. These approaches commonly operate under the assumption that the adversary possesses partial or full knowledge of the victim model's training environment—such as node attributes, the overall graph topology, or relevant subgraph structures. Wu et al. (2022) demonstrates a model extraction attack on GNNs by generating legitimate-looking query nodes and leveraging their responses, along with structural knowledge, to reconstruct the target model. The study systematically defines seven threat models reflecting different levels of adversarial knowledge and introduces tailored attack strategies for each setting. More recent efforts have shifted toward studying MEAs in the **inductive** graph learning setting. Shen et al. (2022) introduced the first set of model stealing attacks targeting inductive GNNs, establishing a formal threat model and designing six distinct attack strategies based on varying levels of adversarial knowledge and the nature of the victim model's responses. Building on this, STEALGNN Zhuang et al. (2024) advances the field by removing the reliance on node features drawn from the same distribution as the victim's training data. Their work presents the first data-free model extraction framework for GNNs, significantly broadening the scope of feasible attack scenarios. Complementary to these MEA studies, recent work shows that GNNs can leak training information in more subtle forms. For example, molecular pre-trained models have been shown to expose training molecules via representation-space consistency Huang et al. (2024), while shared GNNs may also reveal global graph statistics through efficient property inference attacks Yuan et al. (2024). However, none of these works offer theoretical guarantees, and the adversarial formulation of MEAs remains incomplete.

## D.2  LIMITATIONS

While TRIDENT demonstrates significant advancements in data-free model extraction attacks against GNNs, we acknowledge certain limitations and areas for future exploration:

**Theoretical Assumptions and Robustness.** Our theoretical framework, while providing foundational insights, relies on certain assumptions for deriving analytical bounds (concerning aspects like non-zero risk, i.i.d. sampling for target training/test data, and properties of loss functions). For instance, Assumption 1 (Appendix A), which posits that output probabilities are uniformly bounded away from zero, facilitates the analysis of cross-entropy loss but might not hold universally. While the

core principles of minimizing generalization discrepancy are expected to be broadly applicable, the tightness of the derived bounds might vary if these assumptions are significantly violated in highly diverse real-world GMLaaS environments. Future work could explore the robustness of these bounds under a wider range of conditions or relaxed assumptions.

**Attacker-Side Computational Resources.** TRIDENT is designed for query efficiency from the perspective of interacting with the victim model. However, the attack process itself, involving training a graph generator and two surrogate models (main and light), incurs computational costs on the attacker's side. While the light surrogate is designed to be inexpensive, the overall attacker-side overhead, especially when generating and processing very large or complex graphs, was not the primary focus of optimization in this work and could be a consideration for attackers with limited local compute resources.

**On tightness.** Empirically verifying the tightness of lower bound is infeasible in the data-free setting because the bound depends on the victim's true risk over its private data, which is unobservable to the attacker. Instead, we validate the design principle behind the bound, which explicitly minimizes the bound's discrepancy term, attains SOTA fidelity/accuracy under the same budgets.

**On Assumption 1 & i.i.d.** Assumption 1 models the unobservable private training distribution and is necessary to reason about extraction risk. While our analysis assumes i.i.d. sampling within each distribution, our graph datasets are non-i.i.d. in structure; nevertheless, the SOTA performance suggests the minimize-discrepancy principle remains effective beyond the idealized setting.

As an attack framework, TRIDENT's primary contribution is to expose vulnerabilities. The paper focuses on the methodology and effectiveness of the attack itself. A deeper investigation into the downstream privacy implications for the individuals or entities whose data was used to train the original victim GNN, should it be successfully stolen, is outside the current scope but represents an important societal consideration stemming from such attacks.

### D.3 Broader Impacts

**Positive Societal Impacts:** TRIDENT's success in extracting models under strict budget constraints and in data-free settings highlights specific weaknesses that MLaaS providers and GNN developers must address. The unified theoretical account explaining when and why such attacks succeed and the insights into what kinds of queries enable effective MEAs can directly inform the design of more robust defense mechanisms. Although defense strategies such as detection Zhang et al. (2021b); Pal et al. (2021); Liu et al. (2022); Tang et al. (2024); Cheng et al. (2025), watermark Jia et al. (2021); Chakraborty et al. (2022); Lederer et al. (2023), and fingerprinting Waheed et al. (2024a;b); Wu et al. (2024) have been developed, specific defense methods for preventing MEAs still lack of investigation. We expect that future works could extend our adapted label-flipping strategy to provide more insights. As noted in our conclusion, developing theoretically grounded defenses specifically against such theory-aware extraction attacks will be crucial for enhancing the security of MLaaS platforms.

**Negative Societal Impacts:** Despite the intention to improve security, the development and publication of advanced attack methodologies like TRIDENT inherently carry risks if misused. If TRIDENT were to be adopted by malicious actors, it could facilitate the unauthorized replication of proprietary GNN models. This could lead to intellectual property theft, undermining the "commercial interests" of companies that have invested significantly in developing these models. As stated in the introduction, a successfully stolen model could be "exploited for further attacks", potentially leading to secondary privacy breaches if the surrogate model is used to infer sensitive information about the data on which the original victim GNN was trained.

### D.4 Evaluation of Extraction Over Defense

We also implement MEAs on victim model with defense. However, due to the limited prior research on defense for MEAs targeting GNNs, especially in data-free setting, we provide a adapted label-flipping strategies here. Specifically, the adapted defense strategy allows the victim model to keep monitoring the number of queries. If the number of queries exceeds a certain threshold (e.g. 100000 queries hold in Figures), the victim would deliberately output other labels instead of the predicted one. By analyzing Figure 4 and Figure 5, we observe that among all these versions with different budget ratio, the full framework outperforms than others both in the pre-heat stage and alignment

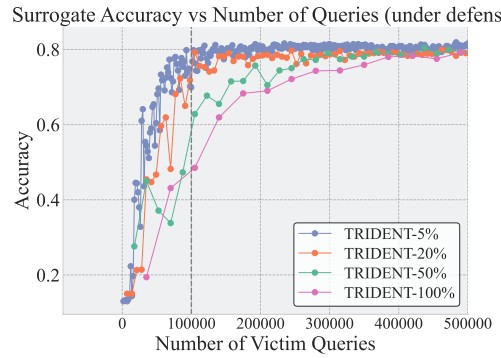 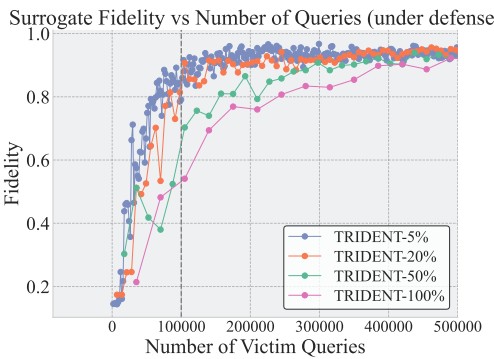

Figure 4: Accuracy of TRIDENT against the label-flipping defense.

Figure 5: Fidelity of TRIDENT against the label-flipping defense.

stage. Additionally, the full version behaves the most robust against such defense, finally achieving almost the same level of accuracy and fidelity. We expect to report the performance of other baselines against such defense in supplementary materials.

### D.5 EVALUATION OF LINK PREDICTION

In addition to node classification, we further evaluate TRIDENT on the link prediction task, which is another widely-used graph learning paradigm in real-world GMLaaS systems. Following the setting in prior data-free MEA literature Zhuang et al. (2024) the attacker queries the victim GNN on induced subgraphs and attempts to reconstruct a surrogate model that approximates the victim's scoring function over node pairs. Specifically, given a graph $G = (V, E)$, link prediction aims to estimate the probability that an edge exists between a node pair $(u, v)$. A victim model $M$ outputs a score $M(u, v) \in [0, 1]$ for each queried pair. The attacker trains a surrogate $M'$ using these scores (or labels), with the goal of minimizing extraction risk under the same black-box, data-free constraints described in Section 5.

Still, we follow the node-level MEA setting, and we generate synthetic graphs via the GraphVAE-style generator and randomly sample node pairs from the induced subgraph $G'_i$. Each queried pair $(u, v)$ is sent to the victim model to obtain its predicted link probability. These labels are used to update both the surrogate and the generator. We evaluate TRIDENT under the same 120-query budget as in the node classification experiments. We use the accuracy and fidelity for metrics, and Cora, CiteSeer, PubMed, and Amazon-Computers for datasets, following the same preprocessing as Section 5. OGB-Arxiv is not reported due to OOM issues. Table7, report link prediction accuracy and fidelity on three backbones. TRIDENT consistently outperforms all baselines across all datasets, demonstrating its superiority under the strict data-free and low-query regime. The improvement is significant on Cora and CiteSeer, where TRIDENT's refined query scheduling enables more informative subgraph synthesis and thus stronger extraction fidelity.

### D.6 RUNNING TIME AND COMPLEXITY

We report the empirical running time of TRIDENT and compare it with prior data-free attacks, followed by a detailed complexity analysis. All methods are evaluated under a fixed query budget of 120 queries. Experiments are conducted on the same hardware environment for fairness.

**Running Time.** Table 8 summarizes the mean running time (in seconds) over multiple runs. As expected, TRIDENT introduces more local optimization steps due to its surrogate refinement and the additional light surrogate, resulting in higher local computation time. However, this trade-off is fundamentally aligned with real MLaaS settings and the time is basically within 2 or 3 mins. API queries incur network latency, rate limits, and monetary charges, whereas local computation

Table 7: Link prediction attack performance of TRIDENT and baselines on GCN.

| Model | Method | Cora | | CiteSeer | | PubMed | | A-Computers | |
|-------|--------|------|-----|----------|-----|--------|-----|-------------|-----|
| | | Acc | Fid | Acc | Fid | Acc | Fid | Acc | Fid |
| GCN | Random Graph | 77.16 | 78.89 | 51.92 | 56.63 | 53.54 | 51.24 | 68.94 | 70.21 |
| GCN | Type III (cos) | 85.58 | 88.96 | 57.74 | 73.35 | 86.02 | 94.52 | 82.40 | 87.74 |
| GCN | Type III (full) | 87.51 | 90.07 | 58.06 | 73.64 | 84.13 | 91.43 | 84.29 | 90.22 |
| GCN | **TRIDENT** | 91.41 | 93.85 | 62.55 | 79.67 | 86.77 | 94.17 | 83.17 | 89.73 |

on the attacker's side is essentially free. TRIDENT deliberately trades extra local computation for significantly stronger extraction under the same number of queries, which is the primary resource constraint in practical black-box attacks.

Table 8: Running time (seconds) under 120-query budget. "1, 2, 3" denote three baseline attacks.

| Model | Dataset | Attack | Mean (s) | Std (s) |
|-------|---------|--------|----------|---------|
| GCN | Citeseer | 1 | 9.98 | 1.25 |
| GCN | Citeseer | 2 | 64.58 | 0.71 |
| GCN | Citeseer | 3 | 17.64 | 0.95 |
| GCN | Citeseer | Trident | 146.47 | 36.68 |
| GCN | A-Computers | 1 | 12.32 | 0.92 |
| GCN | A-Computers | 2 | 26.73 | 0.78 |
| GCN | A-Computers | 3 | 15.75 | 0.92 |
| GCN | A-Computers | Trident | 138.19 | 58.21 |
| GCN | Cora | 1 | 5.64 | 0.78 |
| GCN | Cora | 2 | 30.23 | 1.29 |
| GCN | Cora | 3 | 9.66 | 0.12 |
| GCN | Cora | Trident | 98.09 | 17.09 |
| GCN | Ogb-Arxiv | 1 | 6.75 | 3.93 |
| GCN | Ogb-Arxiv | 2 | 37.61 | 0.16 |
| GCN | Ogb-Arxiv | 3 | 17.44 | 0.00 |
| GCN | Ogb-Arxiv | Trident | 158.39 | 41.88 |
| GCN | Pubmed | 1 | 20.79 | 0.52 |
| GCN | Pubmed | 2 | 30.94 | 0.41 |
| GCN | Pubmed | 3 | 23.58 | 1.16 |
| GCN | Pubmed | Trident | 102.64 | 0.56 |

**Complexity Analysis.** We briefly analyze the computational complexity of TRIDENT and compare it to single-surrogate data-free attacks. Let $B$ be the total query budget, $\alpha$ be the budget ratio, $n_q$ the number of queried nodes per iteration (so $\sum_i n_q = B$), $|E_q|$ the induced edges, and $d$ the node feature dimension. The generator cost is denoted $C_{\text{gen}}(n_q, d)$. For message-passing GNNs (GCN/GAT/SAGE), a forward–backward pass on a graph $(n_q, |E_q|)$ scales as:

$$\tilde{O}\big(L \cdot (|E_q| + n_q) \cdot d\big),$$

where $\tilde{O}$ hides constants. A single TRIDENT iteration consists of, Graph generation by $O(C_{\text{gen}}(n_q, d)) \approx O(n_q d^2 + n_q^2 d)$, Main surrogate update by $O(L_s(|E_q| + n_q)d)$, Light surrogate update by $O(L_\ell(|E_q| + n_q)d)$, where $L_\ell \ll L_s$, and KL-weight computation by $O(n_q d)$.

Thus the per-iteration cost is:

$$T_{\text{iter}} = O(C_{\text{gen}}(n_q, d)) + O((L_s + L_\ell)(|E_q| + n_q)d) + O(n_q d).$$

TRIDENT uses $\alpha$ to increase the number of refinement steps while keeping total queries fixed:

$$\sum_{i=1}^{\alpha t} n_q \approx B.$$

The total local computation is then:

$$T_{\text{TRIDENT}} = \sum_{i=1}^{\alpha t} T_{\text{iter}}^{(i)} = O\Big(\alpha\, C_{\text{gen}}(\bar{n}_q, d) + \alpha(L_s + L_\ell)(\overline{|E_q|} + \bar{n}_q)d\Big).$$

Compared to a single-surrogate baseline, TRIDENT adds a constant-factor overhead from the light surrogate and additional refinement iterations. In real MLaaS environments, the dominant cost is the query budget. API calls are slow, rate-limited, and often monetized; network latency alone may take seconds. In contrast, local GPU/CPU computation is inexpensive for the attacker. TRIDENT is thus designed to trade additional local computation, which scales linearly with $\alpha$ for significantly stronger extraction performance under fixed and limited query access. The higher local running time is a deliberate and practical design choice, consistent with the threat model and the optimization objective.

### D.7 EMPIRICAL EVALUATION OF THE TIGHTNESS

We provide the empirical evaluation of the tightness of Theorem 2, which directly builds upon Theorem 1 and Proposition 1 and yields a linear relation that is more straightforward to validate in practice. As discussed above, the victim's true risk is not observable to the attacker. To approximate it, we follow standard practice and use the performance on an independent test set $T'$ as an unbiased proxy. Specifically, we take Cora with GCN backbone as an example. Firstly we train a victim model following the setting in Section 5.1, and then apply the full-version TRIDENT attack. During the incremental extraction of the TRIDENT attack, say $t = 1, \cdots, K$, we take the surrogate model $h_t$ at time step $t$. For each queried node in $Q^{(t)}$, we compute the empirical risk by

$$R_Q(h_t) = \frac{1}{|Q^{(t)}|} \sum_{x \in Q^{(t)}} \ell(h_t(x), y).$$

On the test set $T'$, for each sample $x$, we have the output $h_T(x)$ and label $y$, we then compute

$$\tilde{R}_T(h_t) = \frac{1}{|T'|} \sum_{x \in T'} \ell(h_t(x), h_T(x)),$$

$$R_T(h_t) = \frac{1}{|T'|} \sum_{(x,y) \in T'} \ell(h_t(x), y),$$

and obtain the victim's training loss $R_T(h_T)$

$$R_T(h_T) = \frac{1}{|T_{\text{train}}|} \sum_{(x,y) \in T_{\text{train}}} \ell(h_T(x), y).$$

We then consider a empirical lower bound $\text{LB}_t(a, b) = a\hat{R}_T(h_t) + b\hat{\Delta}_t$, where $\hat{R}_T(h_t) = R_T(h_T)$ is a constant, $\hat{\Delta}_t = R_T(h_t) - R_Q(h_t)$, $a, b$ are constants. Since $\hat{R}_T(h_t)$ is fixed, and $\text{LB}_t(a, b)$ changes by $\hat{\Delta}_t$, we provide the figure of $\tilde{R}_T(h_t), \text{LB}_t(a, b)$ in Fig 6. The empirical risk $\tilde{R}_T(h_t)$ increases as $\Delta$ grows and shows a nonlinear shape, while the lower bound $\text{LB}_t(a, b)$ follows a straight line from below, as predicted by the linear form of Theorem 2. When $\Delta$ is small, i.e., the query distribution is closer to the training distribution and the surrogate is well aligned with the victim, $\tilde{R}_T(h_t)$ remains low and the gap to $\text{LB}_t(a, b)$ is small, indicating a tight bound. As $\Delta$ becomes larger and the distributional mismatch increases, the empirical risk grows faster than the linear lower bound and the gap widens accordingly. This behavior is consistent with Corollary 1. Also, the tightness guaranteed by Theorem 2 refers to order-level agreement rather than slope-level matching. Therefore, the goal of the bound is not to reproduce the exact slope of $\tilde{R}_T(h_t)$, but to capture the correct increasing order with respect to $\Delta$ while remaining consistently below the empirical curve.

### D.8 FULL RESULTS FOR RESEARCH QUESTIONS

Please see supplementary materials. Experiments on OGB-Arxiv and A-Photos can be found in Table 9.

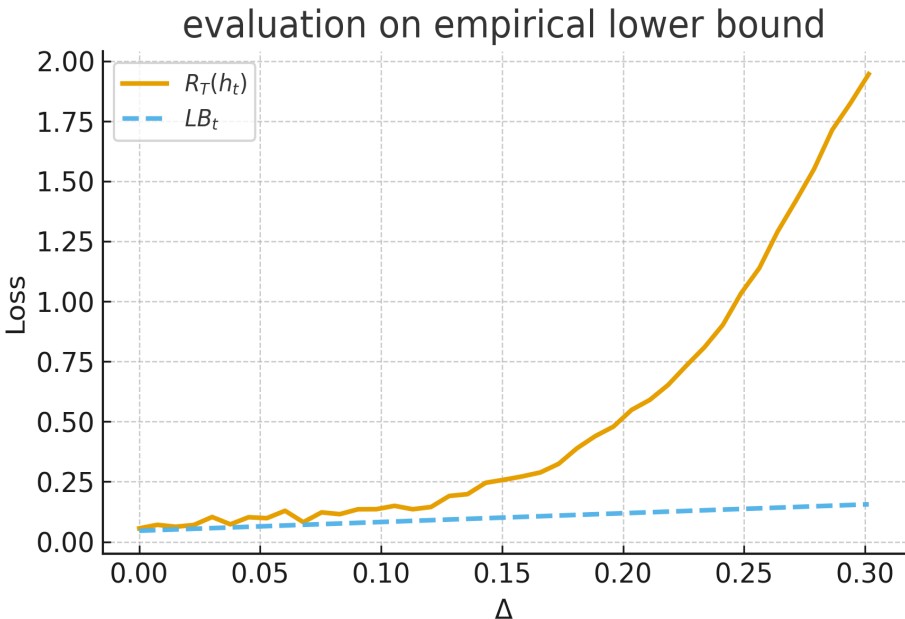

Figure 6: The empirical evaluation of the lower bound.

Table 9: Experimental results for accuracy and fidelity of the surrogate model from attacks.

| Model | Method | A-Photos | | OGB-Arxiv | |
|-------|--------|----------|---|-----------|---|
| | | Accuracy | Fidelity | Accuracy | Fidelity |
| **GAT** | Real Data | $17.70 \pm 6.97$ | $23.02 \pm 4.94$ | $10.82 \pm 1.21$ | $39.68 \pm 2.76$ |
| | Random Graph | $11.21 \pm 3.56$ | $10.23 \pm 2.89$ | $8.72 \pm 0.71$ | $21.17 \pm 2.64$ |
| | Attack III-A | $61.72 \pm 1.24$ | $83.75 \pm 3.00$ | $14.73 \pm 0.53$ | $78.62 \pm 0.47$ |
| | Attack III-F | $63.15 \pm 2.23$ | $88.61 \pm 2.15$ | $13.71 \pm 0.37$ | $77.31 \pm 0.64$ |
| | TRIDENT | $\mathbf{72.78 \pm 0.20}$ | $\mathbf{90.32 \pm 0.81}$ | $\mathbf{30.29 \pm 0.19}$ | $\mathbf{99.23 \pm 0.27}$ |
| **GCN** | Real Data | $18.92 \pm 4.53$ | $24.29 \pm 2.07$ | $9.17 \pm 0.72$ | $20.49 \pm 1.29$ |
| | Random Graph | $13.41 \pm 2.71$ | $10.57 \pm 1.83$ | $7.41 \pm 0.21$ | $12.33 \pm 1.74$ |
| | Attack III-A | $54.31 \pm 0.37$ | $89.93 \pm 1.72$ | $15.31 \pm 0.59$ | $68.29 \pm 3.82$ |
| | Attack III-F | $57.21 \pm 2.45$ | $87.14 \pm 2.42$ | $17.28 \pm 0.42$ | $72.51 \pm 3.71$ |
| | TRIDENT | $\mathbf{72.80 \pm 0.60}$ | $\mathbf{88.73 \pm 0.83}$ | $\mathbf{29.41 \pm 0.52}$ | $\mathbf{99.23 \pm 0.24}$ |
| **SAGE** | Real Data | $23.15 \pm 3.42$ | $23.23 \pm 5.66$ | $10.85 \pm 1.17$ | $18.62 \pm 2.79$ |
| | Random Graph | $11.89 \pm 3.75$ | $18.20 \pm 4.12$ | $7.51 \pm 0.69$ | $16.75 \pm 3.12$ |
| | Attack III-A | $63.82 \pm 5.79$ | $80.80 \pm 6.65$ | $8.72 \pm 0.61$ | $55.21 \pm 1.43$ |
| | Attack III-F | $59.21 \pm 3.85$ | $85.79 \pm 4.11$ | $10.42 \pm 0.81$ | $53.53 \pm 1.25$ |
| | TRIDENT | $\mathbf{75.92 \pm 0.68}$ | $\mathbf{87.52 \pm 2.21}$ | $\mathbf{27.37 \pm 0.33}$ | $\mathbf{99.23 \pm 0.21}$ |

### D.9 DISCLOSURE OF LARGE LANGUAGE MODEL USAGE

During the preparation of this paper, we used LLM-based tools only for grammar checking and minor language polishing. Specifically, some sentences in the Introduction section and a few sentences elsewhere were revised for clarity and grammar.

