Table 1: Experimental results for accuracy and fidelity of the surrogate model from attacks.

| Model | Method | A-Photos | | OGB-Arxiv | |
|---|---|---|---|---|---|
| | | Accuracy | Fidelity | Accuracy | Fidelity |
| GAT | Real Data | $17.70 \pm 6.97$ | $23.02 \pm 4.94$ | $10.82 \pm 1.21$ | $39.68 \pm 2.76$ |
| | Random Graph | $11.21 \pm 3.56$ | $10.23 \pm 2.89$ | $8.72 \pm 0.71$ | $21.17 \pm 2.64$ |
| | Attack III-A | $61.72 \pm 1.24$ | $83.75 \pm 3.00$ | $14.73 \pm 0.53$ | $78.62 \pm 0.47$ |
| | Attack III-F | $63.15 \pm 2.23$ | $88.61 \pm 2.15$ | $13.71 \pm 0.37$ | $77.31 \pm 0.64$ |
| | TRIDENT | $\mathbf{72.78 \pm 0.20}$ | $\mathbf{90.32 \pm 0.81}$ | $\mathbf{30.29 \pm 0.19}$ | $\mathbf{99.23 \pm 0.27}$ |
| GCN | Real Data | $18.92 \pm 4.53$ | $24.29 \pm 2.07$ | $9.17 \pm 0.72$ | $20.49 \pm 1.29$ |
| | Random Graph | $13.41 \pm 2.71$ | $10.57 \pm 1.83$ | $7.41 \pm 0.21$ | $12.33 \pm 1.74$ |
| | Attack III-A | $54.31 \pm 0.37$ | $89.93 \pm 1.72$ | $15.31 \pm 0.59$ | $68.29 \pm 3.82$ |
| | Attack III-F | $57.21 \pm 2.45$ | $87.14 \pm 2.42$ | $17.28 \pm 0.42$ | $72.51 \pm 3.71$ |
| | TRIDENT | $\mathbf{72.80 \pm 0.60}$ | $\mathbf{88.73 \pm 0.83}$ | $\mathbf{29.41 \pm 0.52}$ | $\mathbf{99.23 \pm 0.24}$ |
| SAGE | Real Data | $23.15 \pm 3.42$ | $23.23 \pm 5.66$ | $10.85 \pm 1.17$ | $18.62 \pm 2.79$ |
| | Random Graph | $11.89 \pm 3.75$ | $18.20 \pm 4.12$ | $7.51 \pm 0.69$ | $16.75 \pm 3.12$ |
| | Attack III-A | $63.82 \pm 5.79$ | $80.80 \pm 6.65$ | $8.72 \pm 0.61$ | $55.21 \pm 1.43$ |
| | Attack III-F | $59.21 \pm 3.85$ | $85.79 \pm 4.11$ | $10.42 \pm 0.81$ | $53.53 \pm 1.25$ |
| | TRIDENT | $\mathbf{75.92 \pm 0.68}$ | $\mathbf{87.52 \pm 2.21}$ | $\mathbf{27.37 \pm 0.33}$ | $\mathbf{99.23 \pm 0.21}$ |

# A Technical Appendices and Supplementary Material

## A.1 Evaluation of Extraction Performance

In this subsection, we present additional experimental results comparing the best performance of baselines and TRIDENT. Specifically, we report results on A-Photos and OGB-Arxiv in Table 1, with all other settings being consistent with the main text. The results of TRIDENT are highlighted in bold. As outlined in the experimental setup, we primarily follow prior work, which introduces real data, random graphs, and Type III attacks with two variants. For clarity, we append the suffixes "-A" and "-F" to the attack variant names in Table 1 to distinguish between the two. To evaluate the best extraction performance for each attack type, we set the query budget sufficiently large to ensure peak performance. The best attack performance is determined by the mean fidelity or accuracy over given queries, where further increases in query count do not yield significant improvement. According to Table 1, TRIDENT achieves near state-of-the-art performance across all baselines in both accuracy and fidelity. Notably, while surrogate models on OGB-Arxiv exhibit low accuracy, their high fidelity confirms TRIDENT's superiority over baselines.

## A.2 Evaluation of Query Efficiency

In this subsection, we present experimental results assessing query efficiency. We evaluate performance on CiteSeer, A-Computers, Cora, OGB-Arxiv, and PubMed using GAT, GCN, and SAGE backbones. Additionally, we report the number of queries required for the surrogate model to achieve target fidelity in Tables 2, 3, and 4. Firstly, for a given number of queries, we present a query-fidelity figure to show the efficiency. Secondly, for a certain fidelity, we report the number of queries that the surrogate model needed to achieve it. We record the mean and std value of the first 10 queries for each attack when its fidelity exceeds a threshold. We note that we present N/A here since the two methods cannot achieve the given fidelity for even infinite queries. By presenting these in Figures and Tables, we derive the following key observations: (1) For a given number of queries, TRIDENT achieves not only the highest performance, but also presents a high level of efficiency. For the first few queries, TRIDENT is the quickest to achieve a high performance. (2) For a certain fidelity, TRIDENT requires far fewer number of victim queries to achieve the given fidelity than other baselines. These two observations conclude that TRIDENT is the most efficient attack among the baselines.

Table 2: Number of queries (for given fidelity) of TRIDENT and baselines over all datasets on GAT. The best results are highlighted in bold. Models fail to meet the specified fidelity are marked as N/A.

| Model | Cora ($\times 10^4$) | Citeseer ($\times 10^5$) | PubMed ($\times 10^3$) | A-Computers ($\times 10^5$) | A-Photos ($\times 10^4$) | OGB-Arxiv ($\times 10^2$) |
|---|---|---|---|---|---|---|
| Real Data | N/A | N/A | N/A | N/A | N/A | N/A |
| Random Graph | N/A | N/A | N/A | N/A | N/A | N/A |
| Attack III-A | $14.40 \pm 3.75$ | $3.80 \pm 0.52$ | $3.42 \pm 0.68$ | $4.70 \pm 0.22$ | $10.21 \pm 1.47$ | $8.23 \pm 2.11$ |
| Attack III-F | $15.54 \pm 2.74$ | $3.74 \pm 0.16$ | $4.45 \pm 0.12$ | $6.10 \pm 0.24$ | $9.23 \pm 2.21$ | $7.92 \pm 2.47$ |
| TRIDENT | $\mathbf{6.65 \pm 0.62}$ | $\mathbf{1.12 \pm 0.11}$ | $\mathbf{2.35 \pm 0.37}$ | $\mathbf{2.50 \pm 0.12}$ | $\mathbf{7.57 \pm 1.46}$ | $\mathbf{5.27 \pm 1.23}$ |

Table 3: Number of queries (for given fidelity) of TRIDENT and baselines over all datasets on GCN. The best results are highlighted in bold. Models fail to meet the specified fidelity are marked as N/A.

| Model | Cora ($\times 10^4$) | Citeseer ($\times 10^5$) | PubMed ($\times 10^3$) | A-Computers ($\times 10^5$) | A-Photos ($\times 10^3$) | OGB-Arxiv ($\times 10^2$) |
|---|---|---|---|---|---|---|
| Real Data | N/A | N/A | N/A | N/A | N/A | N/A |
| Random Graph | N/A | N/A | N/A | N/A | N/A | N/A |
| Attack III-A | $15.3 \pm 4.12$ | $4.12 \pm 0.57$ | $4.03 \pm 0.43$ | $5.22 \pm 0.41$ | $8.54 \pm 1.09$ | $6.48 \pm 2.27$ |
| Attack III-F | $17.23 \pm 3.44$ | $4.21 \pm 0.36$ | $4.36 \pm 0.32$ | $6.47 \pm 0.54$ | $9.21 \pm 1.29$ | $5.57 \pm 1.89$ |
| TRIDENT | $\mathbf{6.89 \pm 0.59}$ | $\mathbf{2.82 \pm 0.24}$ | $\mathbf{3.17 \pm 0.54}$ | $\mathbf{3.19 \pm 0.32}$ | $\mathbf{5.47 \pm 1.02}$ | $\mathbf{4.21 \pm 1.19}$ |

## A.3 Evaluation of Ablation Study

In this subsection, we present full ablation study results, comparing the performance of the complete TRIDENT framework against its ablated variants to assess individual component contributions. Experiments were conducted on Cora, CiteSeer, A-Computers, and A-Photos using GAT, GCN, and SAGE backbones. We observe that for the most cases, the budget ratio and the light surrogate model contribute significantly to the performance of TRIDENT. As for the A-Photos dataset, our choice of budget ratio seems not to outperform others, while the intermediate version, $\alpha = 2$ (i.e., TRIDENT-50%) still outperforms the ablated version (i.e., TRIDENT-100%). This reveals that the introduction of budget ratio does help TRIDENT to better implement attacks. The following key observations are drawn from this evaluation: (1) The incorporation of the budget ratio significantly improves both the quality and efficiency to TRIDENT. Regarding quality, we observe that a higher budget ratio always leads to better fidelity performance. Regarding efficiency, for given number of victim queries (e.g. $200,000$), the ablated version (i.e., TRIDENT-100%) yields the lowest fidelity among all the versions. (2) The light surrogate model also contributes to the efficiency to TRIDENT, and slightly increases the best accuracy and fidelity that can be achieved. This observation agrees with our goal that the light surrogate model is designed for a cost-efficient and query-free refinement, while the budget ratio provides the primary improvement in efficiency and performance.

## A.4 Evaluation of Extraction Over Defense

In this subsection, we present results for MEAs on victim models with defense mechanisms. We adopt the same defense strategy as in the main text, evaluating TRIDENT's performance with varying budget ratios on Cora, CiteSeer, PubMed, A-Photos, and A-Computers. We note defense strategies are implemented only on GCN backbones, as GAT and SAGE may not support this defense type. Discussion of defense applicability across backbones is beyond this paper's scope. Specifically, the adapted defense strategy allows the victim model to keep monitoring the number of queries. If the number of queries exceeds a certain threshold (e.g. $100,000$ queries hold in Figures), the victim would deliberately output other labels instead of the predicted one. By analyzing Figures, we observe that among all these versions with different budget ratio, the full framework outperforms than others both in the pre-heat stage and alignment stage. Additionally, the full version behaves the most robust against such defense, finally achieving almost the same level of accuracy and fidelity.

Table 4: Number of queries (for given fidelity) of TRIDENT and baselines over all datasets on SAGE. The best results are highlighted in bold. Models fail to meet the specified fidelity are marked as N/A.

| Model | Cora ($\times 10^4$) | Citeseer ($\times 10^5$) | PubMed ($\times 10^3$) | A-Computers ($\times 10^5$) | A-Photos ($\times 10^4$) | OGB-Arxiv ($\times 10^2$) |
|---|---|---|---|---|---|---|
| Real Data | N/A | N/A | N/A | N/A | N/A | N/A |
| Random Graph | N/A | N/A | N/A | N/A | N/A | N/A |
| Attack III-A | $14.4 \pm 3.75$ | $3.8 \pm 0.52$ | $3.42 \pm 0.68$ | $5.43 \pm 0.56$ | $11.51 \pm 1.24$ | $8.21 \pm 2.37$ |
| Attack III-F | $15.54 \pm 2.74$ | $3.74 \pm 0.16$ | $4.45 \pm 0.12$ | $6.62 \pm 0.71$ | $9.76 \pm 0.92$ | $7.49 \pm 1.75$ |
| TRIDENT | $\mathbf{6.65 \pm 0.62}$ | $\mathbf{1.12 \pm 0.11}$ | $\mathbf{2.35 \pm 0.37}$ | $\mathbf{3.34 \pm 0.45}$ | $\mathbf{7.32 \pm 1.19}$ | $\mathbf{5.15 \pm 0.72}$ |

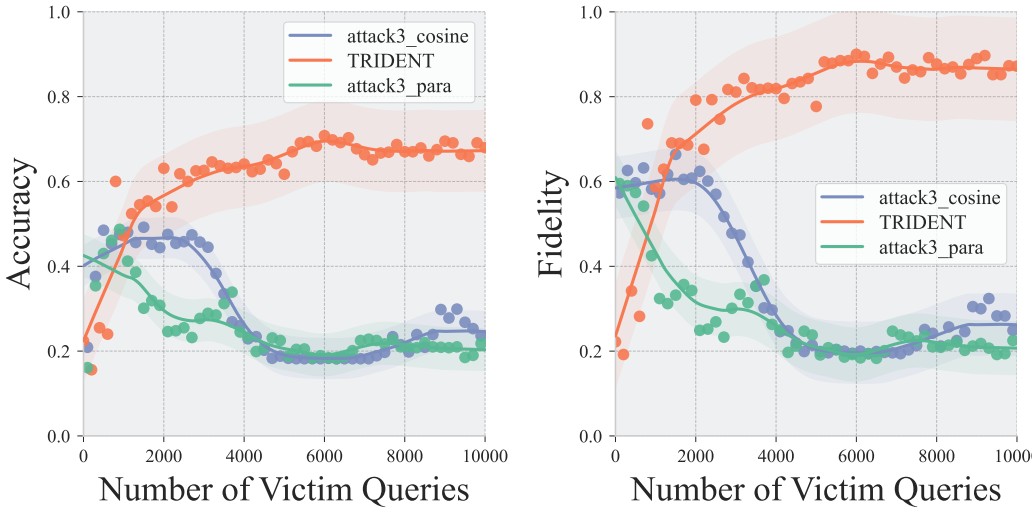

(a) Accuracy on surrogate model over CiteSeer with GAT backbone.

(b) Fidelity on surrogate model over CiteSeer with GAT backbone.

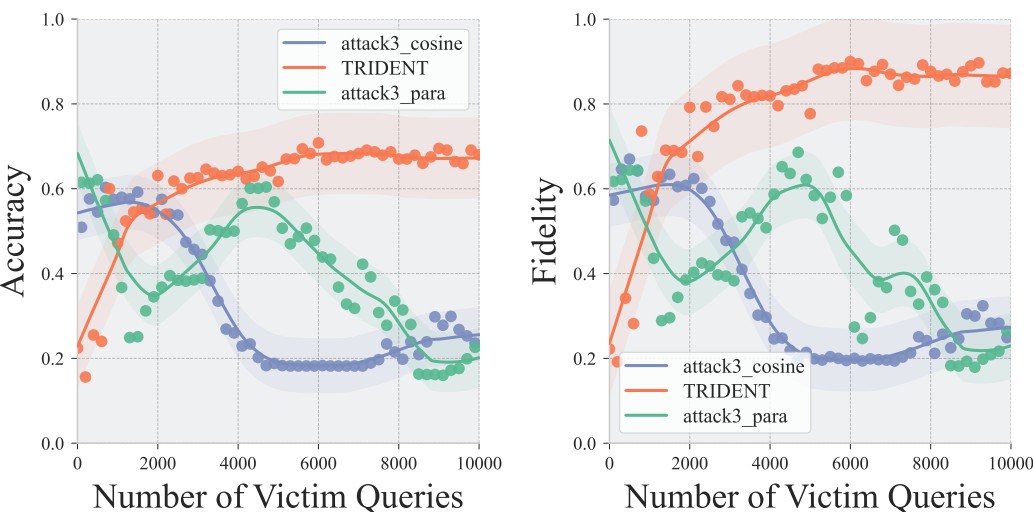

(a) Accuracy on surrogate model over CiteSeer with SAGE backbone.

(b) Fidelity on surrogate model over CiteSeer with SAGE backbone.

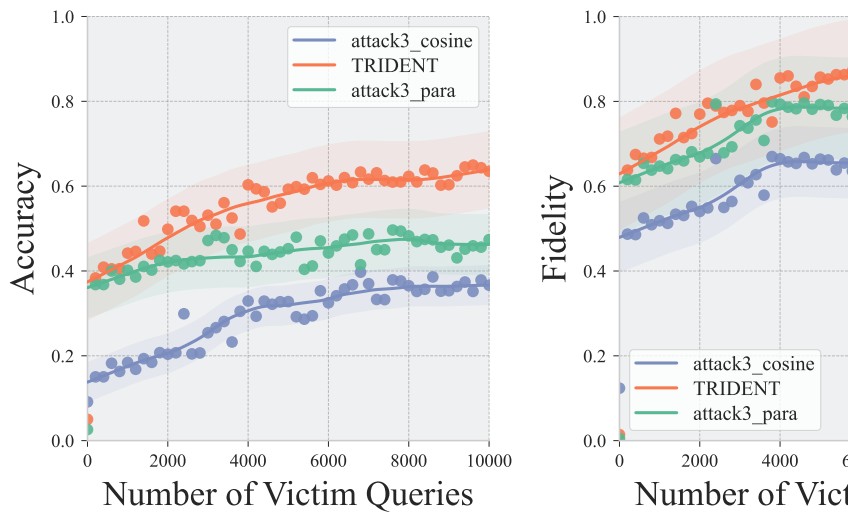

(a) Accuracy on surrogate model over A-Computers with GAT backbone.

(b) Fidelity on surrogate model over A-Computers with GAT backbone.

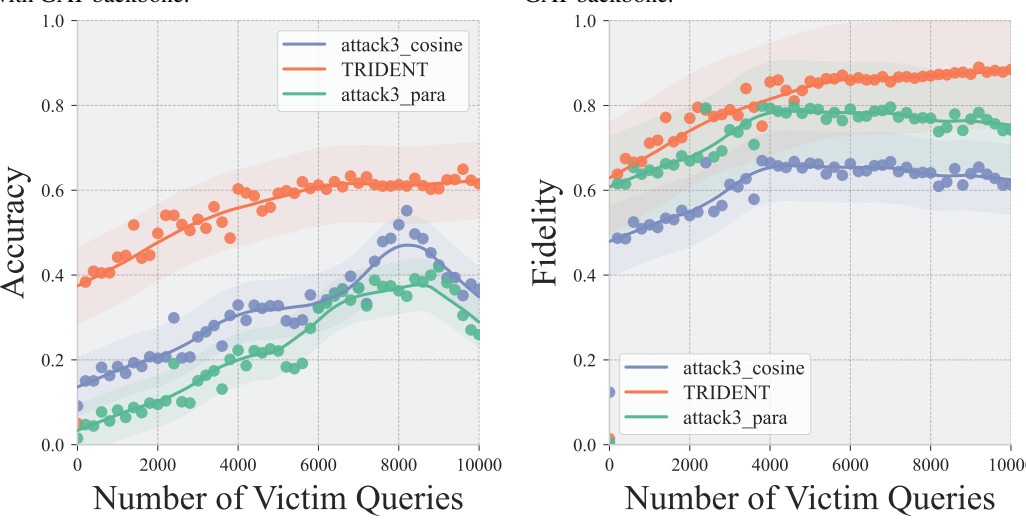

(a) Accuracy on surrogate model over A-Computers with GCN backbone.

(b) Fidelity on surrogate model over A-Computers with GCN backbone.

(a) Accuracy on surrogate model over A-Computers with SAGE backbone.

(b) Fidelity on surrogate model over A-Computers with SAGE backbone.

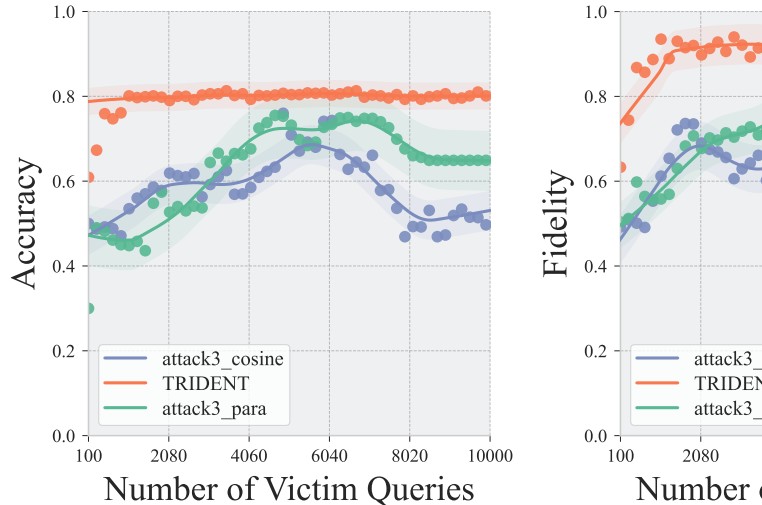
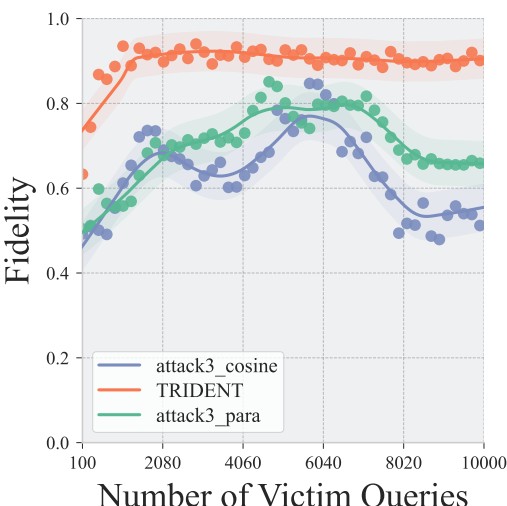

(a) Accuracy on surrogate model over Cora with GAT backbone.

(b) Fidelity on surrogate model over Cora with GAT backbone.

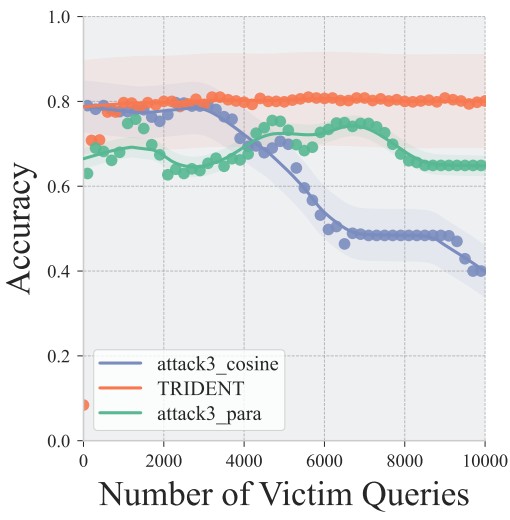

(a) Accuracy on surrogate model over Cora with GCN backbone.

(b) Fidelity on surrogate model over Cora with GCN backbone.

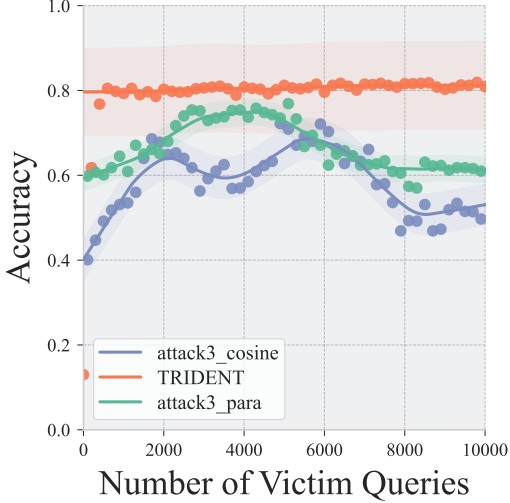
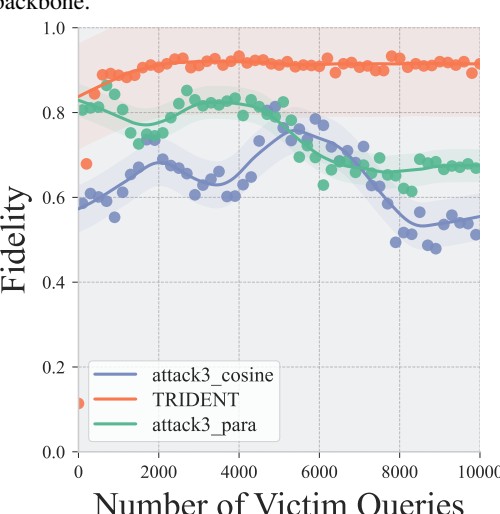

(a) Accuracy on surrogate model over Cora with SAGE backbone.

(b) Fidelity on surrogate model over Cora with SAGE backbone.

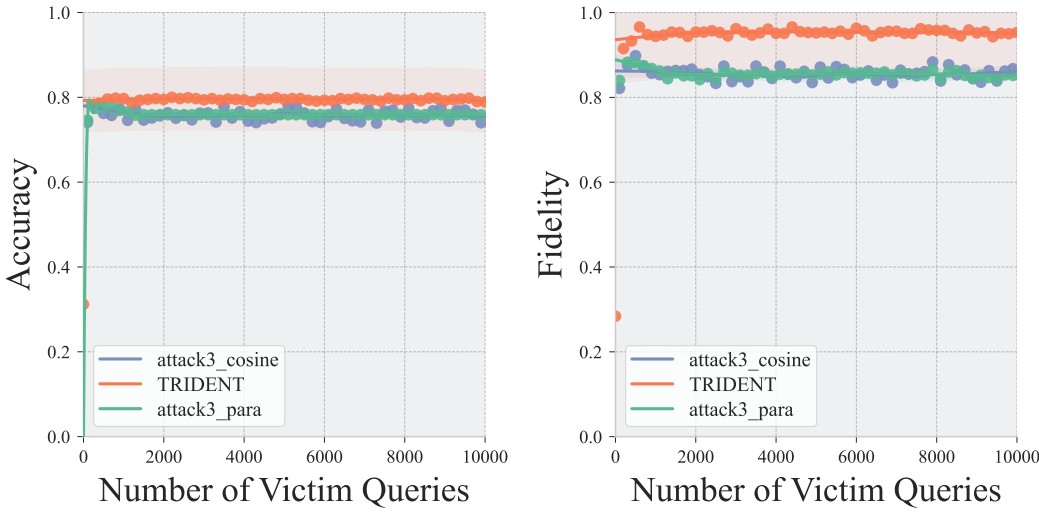

(a) Accuracy on surrogate model over PubMed with GAT backbone.

(b) Fidelity on surrogate model over PubMed with GAT backbone.

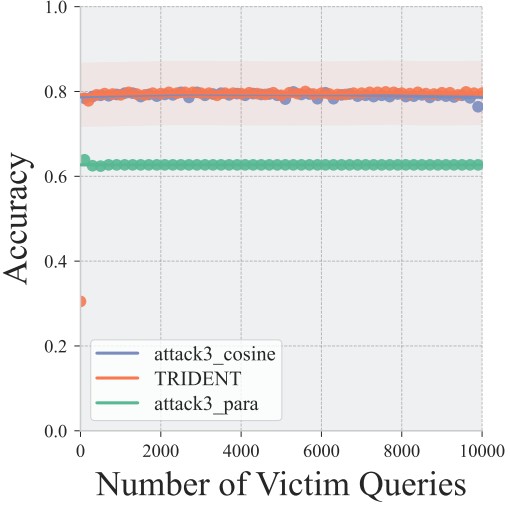

(a) Accuracy on surrogate model over PubMed with GCN backbone.

(b) Fidelity on surrogate model over PubMed with GCN backbone.

(a) Accuracy on surrogate model over PubMed with SAGE backbone.

(b) Fidelity on surrogate model over PubMed with SAGE backbone.

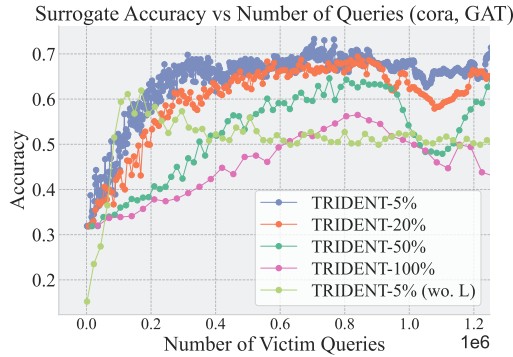
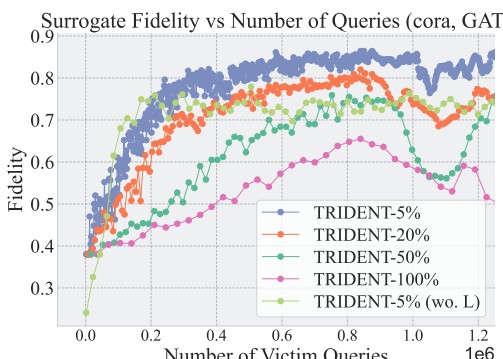

(a) Accuracy on ablated versions over Cora with GAT backbone.

(b) Fidelity on ablated versions over Cora with GAT backbone.

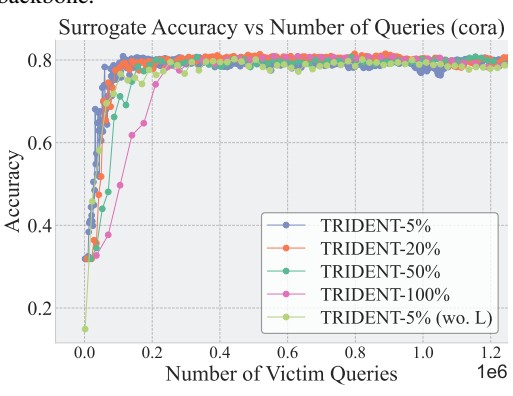
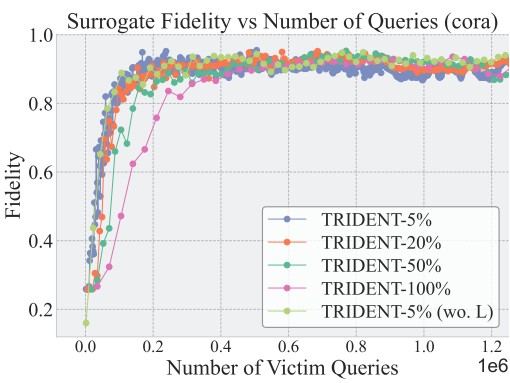

(a) Accuracy on ablated versions over Cora with GCN backbone.

(b) Fidelity on ablated versions over Cora with GCN backbone.

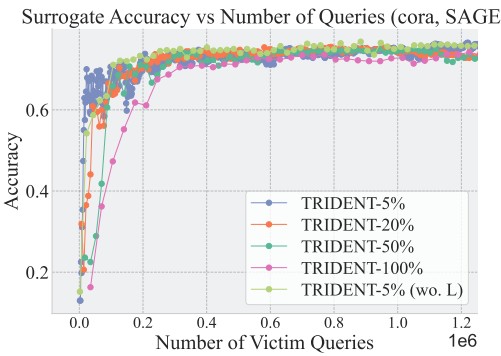
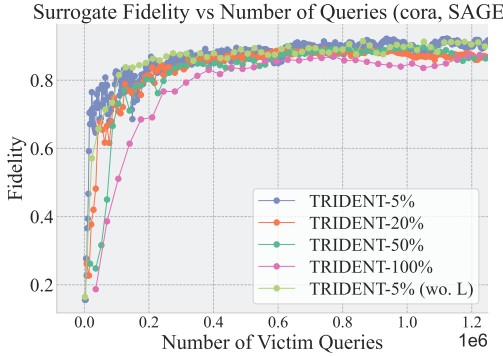

(a) Accuracy on ablated versions over Cora with SAGE backbone.

(b) Fidelity on ablated versions over Cora with SAGE backbone.

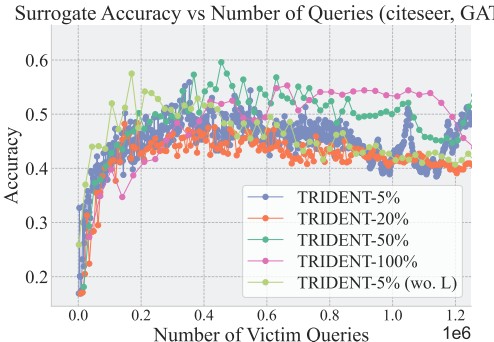
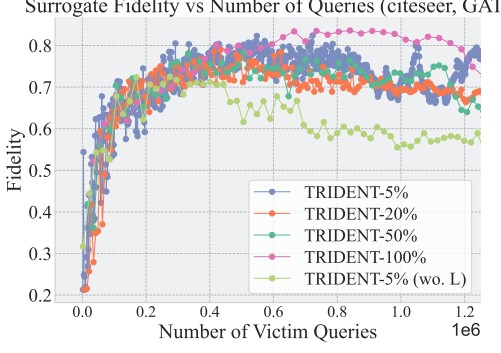

(a) Accuracy on ablated versions over CiteSeer with GAT backbone.

(b) Fidelity on ablated versions over CiteSeer with GAT backbone.

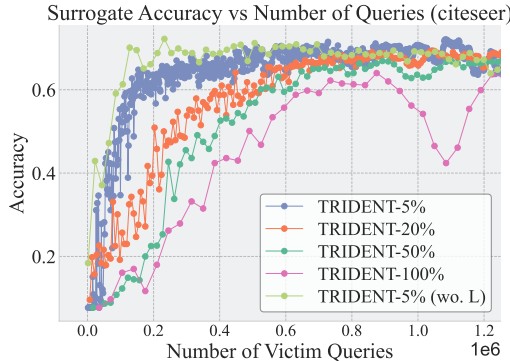

(a) Accuracy on ablated versions over CiteSeer with GCN backbone.

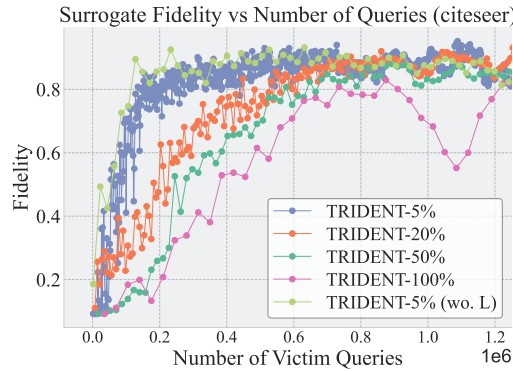

(b) Fidelity on ablated versions over CiteSeer with GCN backbone.

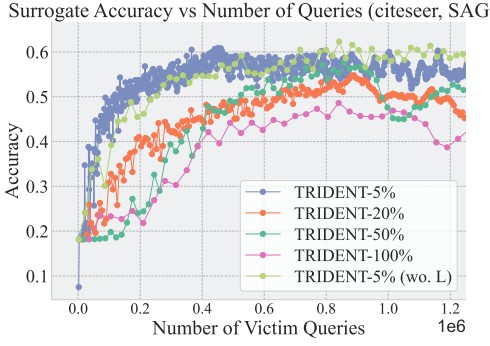

(a) Accuracy on ablated versions over CiteSeer with SAGE backbone.

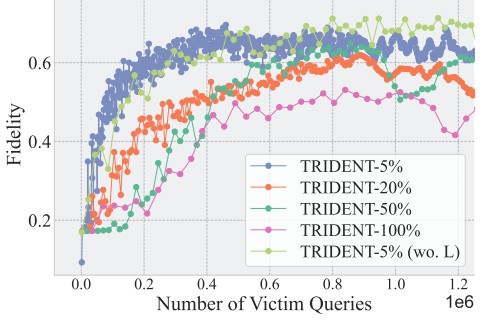

(b) Fidelity on ablated versions over CiteSeer with SAGE backbone.

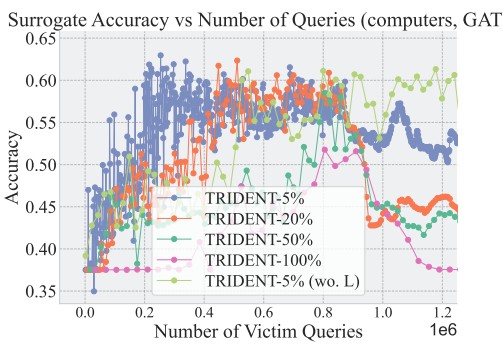

(a) Accuracy on ablated versions over A-Computers with GAT backbone.

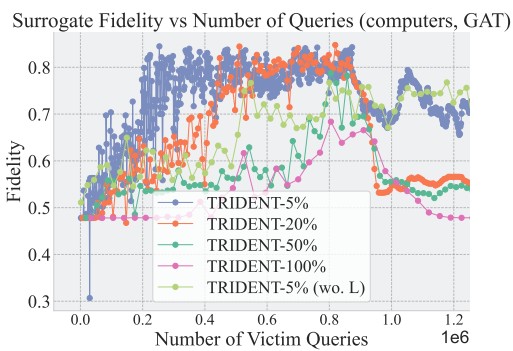

(b) Fidelity on ablated versions over A-Computers with GAT backbone.

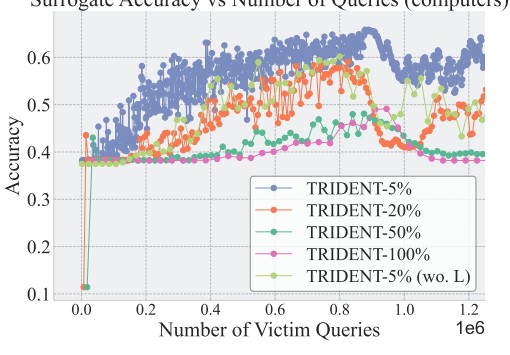

(a) Accuracy on ablated versions over A-Computers with GCN backbone.

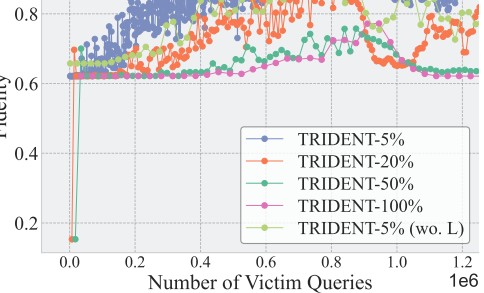

(b) Fidelity on ablated versions over A-Computers with GCN backbone.

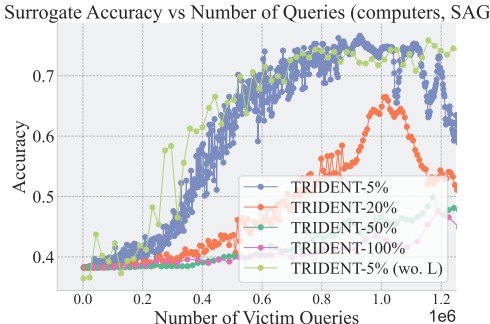

(a) Accuracy on ablated versions over A-Computers with SAGE backbone.

(b) Fidelity on ablated versions over A-Computers with SAGE backbone.

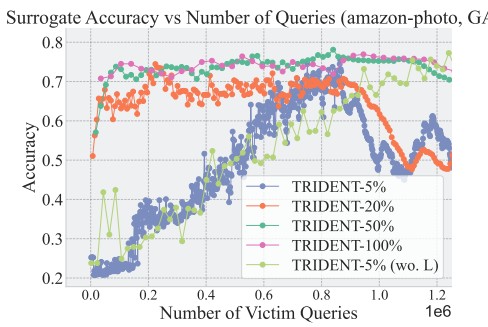
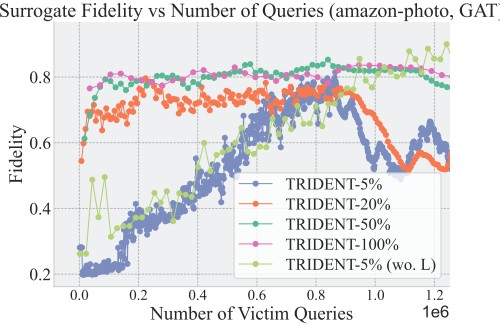

(a) Accuracy on ablated versions over A-Photos with GAT backbone.

(b) Fidelity on ablated versions over A-Photos with GAT backbone.

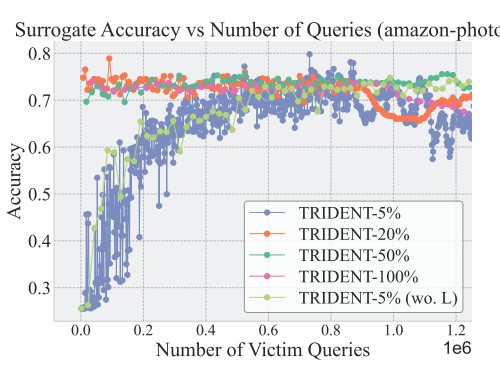
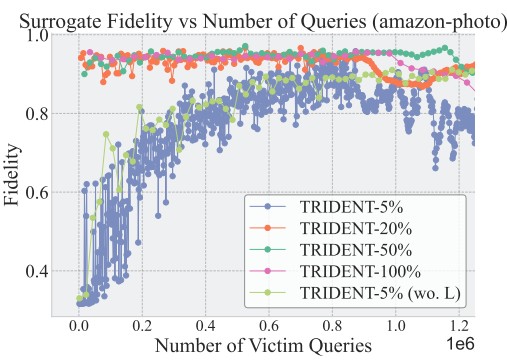

(a) Accuracy on ablated versions over A-Photos with GCN backbone.

(b) Fidelity on ablated versions over A-Photos with GCN backbone.

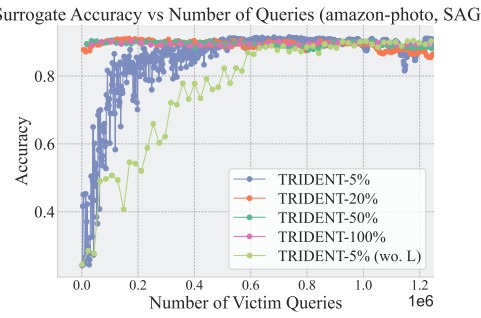

(a) Accuracy on ablated versions over A-Photos with SAGE backbone.

(b) Fidelity on ablated versions over A-Photos with SAGE backbone.

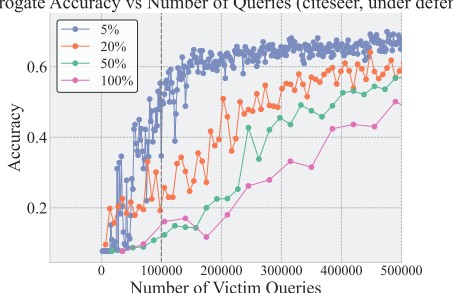

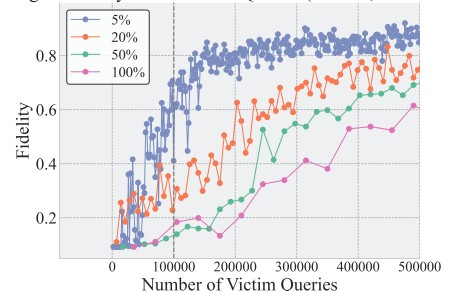

(a) Accuracy of TRIDENT with different budget ratio against the label-flipping defense on CiteSeer.

(b) Fidelity of TRIDENT with different budget ratio against the label-flipping defense on CiteSeer.

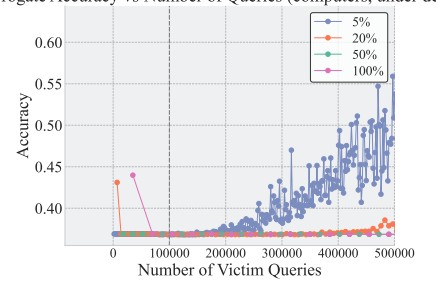

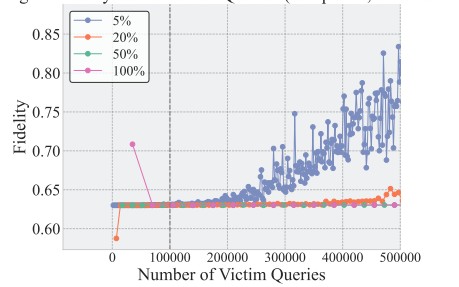

(a) Accuracy of TRIDENT with different budget ratio against the label-flipping defense on A-Computers.

(b) Fidelity of TRIDENT with different budget ratio against the label-flipping defense on A-Computers.

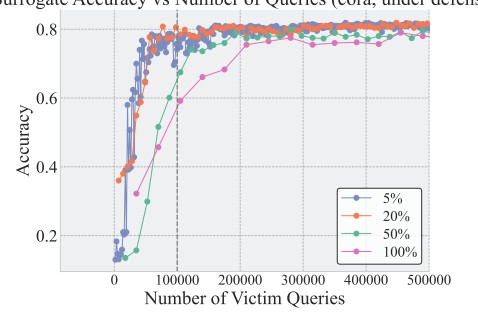

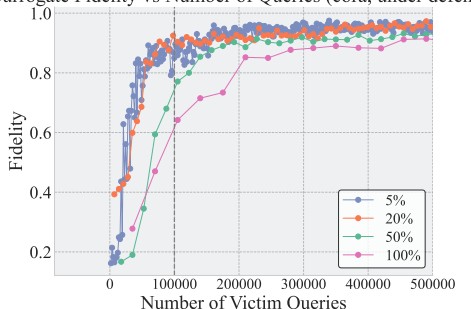

(a) Accuracy of TRIDENT with different budget ratio against the label-flipping defense on Cora.

(b) Fidelity of TRIDENT with different budget ratio against the label-flipping defense on Cora.

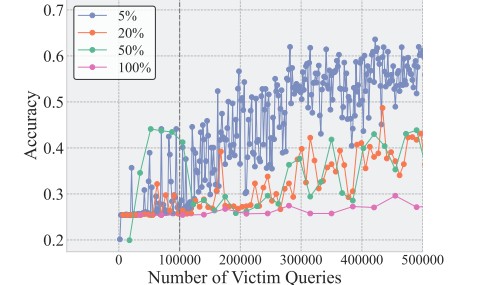

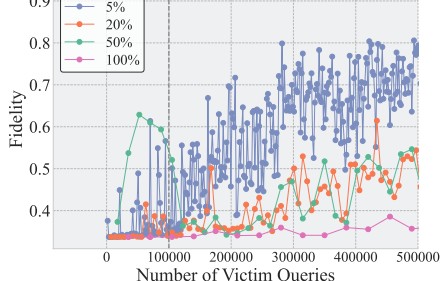

(a) Accuracy of TRIDENT with different budget ratio against the label-flipping defense on A-Photos.

(b) Fidelity of TRIDENT with different budget ratio against the label-flipping defense on A-Photos.

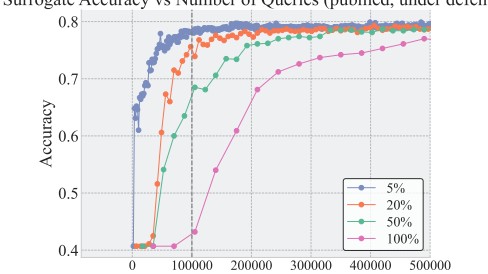
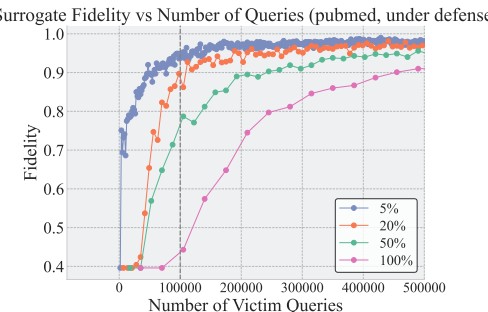

(a) Accuracy of TRIDENT with different budget ratio against the label-flipping defense on PubMed.

(b) Fidelity of TRIDENT with different budget ratio against the label-flipping defense on PubMed.