# OpenReview forum: "TRIDENT: An Efficient Data-Free Model Extraction Attack for Graph Neural Networks"
_ICLR.cc/2026/Conference — Submitted to ICLR 2026_

### Official Review · Reviewer_Peod · 2025-10-27

**Soundness:** 3
**Presentation:** 2
**Contribution:** 3
**Rating:** 6
**Confidence:** 3

**Summary:**

The manuscript presents a theoretical and practical framework for MEAs on GNNs offered via MLaaS APIs. It formalizes extraction risk and derives a lower bound linking it to the victim model’s risk and the distributional discrepancy between the query and target data, revealing that minimizing this discrepancy is key to successful extraction. Building on this insight, the paper introduces TRIDENT, a data-free and query-efficient MEA that strategically schedules queries under strict budget constraints. Extensive experiments on six real-world datasets and three GNN backbones show that TRIDENT achieves state-of-the-art extraction performance.

**Strengths:**

The manuscript makes both theoretical and practical advances in MEAs on GNNs. It introduces a rigorous theoretical framework that formalizes extraction risk and derives a lower bound linking it to the victim’s intrinsic risk and the generalization discrepancy between query and target distributions, offering a clear design principle for attack optimization. Building upon this insight, the proposed TRIDENT method implements a data-free and query-efficient attack through two novel components: a budget-ratio-based graph generator for adaptive query scheduling and a surrogate-as-victim alignment module that refines surrogate learning without additional queries. Theoretical derivations are well integrated with algorithmic design, and extensive experiments across six benchmark datasets and three GNN architectures demonstrate state-of-the-art fidelity and accuracy, validating both the theoretical analysis and empirical effectiveness. The ablation study further substantiates each module’s contribution to performance and efficiency.

**Weaknesses:**

Despite its theoretical elegance and empirical rigor, the manuscript exhibits several limitations in clarity, scope, and practical depth that may restrict its impact and generalizability.

The paper’s theoretical sections are highly abstract and mathematically dense. The presentation relies heavily on many notations and several assumptions, which may obscure the underlying intuition behind the derived results. It may hinder accessibility for readers less familiar with formal risk analysis or statistical learning theory.

Although the proposed TRIDENT framework emphasizes query efficiency, the manuscript does not explicitly analyze the computational or memory overhead introduced by the dual-surrogate design and graph generation proces. For instance, maintaining both the main and light surrogate models, along with iterative graph synthesis and KL-weighted updates, could significantly increase training time and hardware demands. Without explicit runtime or complexity analysis, it remains unclear whether the claimed efficiency gains in query usage translate into practical computational efficiency in real-world MLaaS environments.

The experimental evaluation is confined to node classification benchmarks (e.g., Cora, CiteSeer, PubMed). This narrow focus limits the generality of conclusions regarding other graph learning paradigms such as link prediction, graph classification, or heterogeneous graphs. Moreover, all datasets are publicly available academic benchmarks, which may not fully capture the data complexity or noise characteristics of industrial GMLaaS deployments.

The study primarily compares TRIDENT with prior data-free attacks such as Type III attacks from Zhuang et al. (2024) but omits recent adaptive MEAs or defense-aware extraction techniques (e.g., dynamic watermarking, detection-based defenses, or distributionally equivalent attacks). This limits the ability to contextualize TRIDENT’s robustness under adversarially hardened environments or modern countermeasures.

Several assumptions in the theoretical derivation (e.g., i.i.d. sampling between query and target distributions, bounded losses, Lipschitz continuity) are strong and may not hold in practical black-box MLaaS settings. The manuscript does not investigate how deviations from these assumptions affect the tightness of the lower bound or the validity of the derived conclusions. This weakens the theoretical generality beyond controlled settings.

Although the ablation study is presented, it primarily focuses on validating the functional roles of individual modules. The manuscript, however, does not include parameter sensitivity analyses that examine how variations in key hyperparameters (e.g., the budget ratio, surrogate model size, or KL weighting factor) affect convergence behavior, fidelity performance, and query efficiency. Incorporating such analyses would offer a more comprehensive understanding of TRIDENT’s stability, robustness, and reproducibility under diverse experimental configurations.

**Questions:**

Please refer to the ``Weaknesses`` part.

---

> ### Author Response · Authors · 2025-11-24
>
> Thank you for your valuable feedback and for acknowledging the insights of our proposal. Below are our responses:
>
> ### Response to W1 (Readability)
> Thanks for pointing this out. Our goal is to establish a unified extraction-risk formulation applicable across graph domains, which required certain notations and formal assumptions. We have already provided intuitive explanations for readers with limited theoretical background in the previous version, and **we've reorganized some definitions to emphasize their conceptual roles in Section 2.**
>
> ### Response to W2 (Computational Costs)
> Thank you for your valuable feedback. Although TRIDENT trains a generator and two surrogate models, all components are intentionally lightweight—the light surrogate uses reduced dimensions, and the generator is a compact GraphVAE prior. While this introduces more local computation than single-surrogate baselines, the additional overhead is modest and remains within a few minutes per run. Crucially, in real MLaaS settings, query usage, not local computation, is the primary bottleneck. API queries are slow and often billed, whereas local GPU/CPU cycles for attackers incur negligible cost. Our design focuses on this regime, where the attacker’s primary goal is to achieve the strongest possible extraction under limited query access, rather than minimizing local training time. A runtime table and a detailed complexity analysis have been added to the revised manuscript to highlight this practical trade-off. **Please refer to Appendix D.6** Our empirical results and complexity analysis show that TRIDENT introduces moderate local computation while yielding substantially better extraction under the same query budget, supporting our claim that local optimization is a worthwhile trade-off in realistic MLaaS threat models.
>
> ### Response to W3 (Industrial GMLaaS)
> Thank you for pointing these out. Node prediction is indeed a common task in industrial GMLaaS systems, and due to the limited availability of industrial-scale datasets, our evaluation is conducted on widely used academic benchmarks. **We agree that extending to other graph learning paradigms is valuable, and we have added discussion on edge prediction tasks in Appendix D.5.** The link prediction results show that TRIDENT almost surpasses all baselines under the same data-free setting, demonstrating that our framework naturally extends beyond node classification and remains effective in broader graph learning paradigms.
>
> ### Response to W4 (Assumptions)
> Thanks for your valuable feedback. The assumptions are standard in learning-theoretic analyses and formally stated in Appendix A. Importantly, all components in TRIDENT satisfy these assumptions in practice. This ensures that the derived bound remains non-vacuous under our setting. **We agree that in the real-settings the assumptions may not hold, please refer to our discussion about Limitations in Appendix D.2.**
>
> ### Response to W5 (Parameter Analyses)
> Thank you for this insightful feedback. We appreciate the suggestion regarding hyperparameter sensitivity. We have now added parameter analyses for key components, particularly for the KL weighting factor, since the budget ratio is already discussed in the main text, and the analysis of model size can be found in prior work such as [1]. These additions provide a more comprehensive understanding of convergence behavior and query efficiency. **We've added these in Appendix C.5.** Our parameter study shows that TRIDENT remains stable across a broad range of KL weights, supporting the reliability of our framework under different hyperparameter configurations.
>
> [1] Zhuang, Yuanxin, et al. "Unveiling the Secrets without Data: Can Graph Neural Networks Be Exploited through {Data-Free} Model Extraction Attacks?." *33rd USENIX Security Symposium (USENIX Security 24). 2024*.

---

> ### Author Response · Authors · 2025-11-25
> **Additional experimental results (1/3)**
>
> We thank you again for your valuable comments. Below are our additional experimental results.
> ### Response to W2: Computational Costs - Appendix D.6
> | Model | Dataset | Attack | Mean (s) | Std (s) |
> | :--- | :--- | :--- | :--- | :--- |
> | GCN | Citeseer | 1 | 9.98 | 1.25 |
> | GCN | Citeseer | 2 | 64.58 | 0.71 |
> | GCN | Citeseer | 3 | 17.64 | 0.95 |
> | GCN | Citeseer | Trident | 146.47 | 36.68 |
> | GCN | A-Computers | 1 | 12.32 | 0.92 |
> | GCN | A-Computers | 2 | 26.73 | 0.78 |
> | GCN | A-Computers | 3 | 15.75 | 0.92 |
> | GCN | A-Computers | Trident | 138.19 | 58.21 |
> | GCN | Cora | 1 | 5.64 | 0.78 |
> | GCN | Cora | 2 | 30.23 | 1.29 |
> | GCN | Cora | 3 | 9.66 | 0.12 |
> | GCN | Cora | Trident | 98.09 | 17.09 |
> | GCN | Ogb-Arxiv | 1 | 6.75 | 3.93 |
> | GCN | Ogb-Arxiv | 2 | 37.61 | 0.16 |
> | GCN | Ogb-Arxiv | 3 | 17.44 | 0.00 |
> | GCN | Ogb-Arxiv | Trident | 158.39 | 41.88 |
> | GCN | Pubmed | 1 | 20.79 | 0.52 |
> | GCN | Pubmed | 2 | 30.94 | 0.41 |
> | GCN | Pubmed | 3 | 23.58 | 1.16 |
> | GCN | Pubmed | Trident | 102.64 | 0.56 |

---

> ### Author Response · Authors · 2025-11-25
> **Additional experimental results (2/3)**
>
> ### Response to W3: Industrial GMLaaS / Link Prediction - Appendix D.5
> | Model | Method | Cora (Acc) | Cora (Fid) | CiteSeer (Acc) | CiteSeer (Fid) | PubMed (Acc) | PubMed (Fid) | A-Computers (Acc) | A-Computers (Fid) |
> | :--- | :--- | :--- | :--- | :--- | :--- | :--- | :--- | :--- | :--- |
> | GCN | Random Graph | 77.16 | 78.89 | 51.92 | 56.63 | 53.54 | 51.24 | 68.94 | 70.21 |
> | GCN | Type III (cos) | 85.58 | 88.96 | 57.74 | 73.35 | 86.02 | 94.52 | 82.40 | 87.74 |
> | GCN | Type III (full) | 87.51 | 90.07 | 58.06 | 73.64 | 84.13 | 91.43 | 84.29 | 90.22 |
> | GCN | **TRIDENT** | 91.41 | 93.85 | 62.55 | 79.67 | 86.77 | 94.17 | 83.17 | 89.73 |

---

> ### Author Response · Authors · 2025-11-25
> **Additional experimental results (3/3)**
>
> ### Response to W5: Parameter Analyses - Appendix C.5
> | KL weight | Cora (Acc) | Cora (Fid) | CiteSeer (Acc) | CiteSeer (Fid) | PubMed (Acc) | PubMed (Fid) | A-Computers (Acc) | A-Computers (Fid) |
> | :--- | :--- | :--- | :--- | :--- | :--- | :--- | :--- | :--- |
> | 0.30 | 63.15 | 75.20 | 62.11 | 77.74 | 71.21 | 85.94 | 48.79 | 75.03 |
> | 0.35 | 69.53 | 80.01 | 63.41 | 79.84 | 72.51 | 88.75 | 50.92 | 76.87 |
> | 0.40 | 72.68 | 83.26 | 67.83 | 81.71 | 76.02 | 91.86 | 54.95 | 80.12 |
> | 0.45 | 77.59 | 87.94 | 69.10 | 84.37 | 77.82 | 93.41 | 57.29 | 82.61 |
> | 0.50 | 80.26 | 91.44 | 70.75 | 86.39 | 79.94 | 95.25 | 61.08 | 86.86 |
> | 0.55 | 76.87 | 87.20 | 68.52 | 83.97 | 74.32 | 91.86 | 58.12 | 82.89 |
> | 0.60 | 70.49 | 80.97 | 67.87 | 82.85 | 71.27 | 89.03 | 57.06 | 80.71 |
> | 0.65 | 64.28 | 74.78 | 65.28 | 81.64 | 70.74 | 87.52 | 53.77 | 77.24 |
> | 0.70 | 56.14 | 68.88 | 63.27 | 79.92 | 67.48 | 84.28 | 52.11 | 76.90 |

---

### Official Review · Reviewer_M3Vq · 2025-10-30

**Soundness:** 4
**Presentation:** 2
**Contribution:** 3
**Rating:** 6
**Confidence:** 4

**Summary:**

This paper proposes TRIDENT, with theory-driven, data-free, and query-efficient model extraction attack for GNNs deployed in MLaaS. It formalizes extraction risk and derives a lower bound linking it to the victim model’s inherent risk and generalization discrepancy between query and target distributions, guiding the design principle of minimizing discrepancy. TRIDENT integrates a budget ratio based graph generator and a Surrogate-as-Victim alignment module to optimize query efficiency and surrogate model fidelity. Extensive experiments on six real-world datasets and three GNN backbones demonstrate its state-of-the-art performance in accuracy, and query efficiency compared to existing baselines.

**Strengths:**

1. First provides a unified theoretical framework for GNN MEAs, addressing the lack of rigorous theory in prior empirical works.

2.Proposes a novel and realistic setting, limited queries, black-box access, and no training data—well aligned with real-world MLaaS security scenarios.

3.Presents comprehensive experimental validation across diverse datasets and models.

**Weaknesses:**

1.The paper is overly formalized; the authors may consider simplifying some definitions. For instance, Definition 1 and Definition 4 appear similar, which reduces readability.

2.It would be valuable to discuss possible defense strategies. Under the setting of limited queries, black-box access, and no training data, it seems difficult for pre-trained model owners to mitigate such attacks.

3.Can the tightness of the theoretical lower bound be empirically validated due to the unobservability of the victim’s training data?

4.Under the no-training-data assumption, how is a suitable graph dataset obtained for querying? The initialization of the graph generator seems crucial. Moreover, TRIDENT requires training both a graph generator and two surrogate models, which introduces non-trivial local computational costs.

5.The paper appears to lack discussion of related works on graph model extraction attacks, such as [1].

[1] Extracting Training Data from Molecular Pre-trained Models

**Questions:**

See weakness.

---

> ### Author Response · Authors · 2025-11-24
>
> Thank you for your valuable feedback and for acknowledging the insights of our proposal. Below are our responses:
>
> ### Response to W1 (Readability)
> Thanks for pointing this out. We agree that Definitions 1 and 4 may appear similar at first glance. Their purposes, however, are different. Definition 1 presents a general extraction risk formulation over a measurable input space, whereas Definition 4 instantiates the query distribution discrepancy. **We've revised the notation to reduce redundancy and make their respective roles clearer in Section 2**, specifically, we added a graph-specific description and straghtforward definition for the test risk.
>
> ### Response to W2 (Defense Strategies)
> Thank you for pointing this out. **We have discussed potential defense strategies in Appendix D.4.** The results demonstrate that TRIDENT preserves strong extraction performance under the adapted label-flipping defense. Under our strict setting, limited queries, black-box access, and no training data, we expect many classical defenses become ineffective, which we believe highlights a meaningful direction for future research.
>
> ### Response to W3 (Tightness of the Bounds)
> Thank you for your valuable feedback. As discussed in Proposition 1 and Appendix A, the tightness of the lower bound relies on standard assumptions such as bounded inputs, Lipschitz losses, etc. TRIDENT satisfies all these assumptions, ensuring that the constants remain finite and the bound is non-vacuous. We will clarify this and add an experiment to show tightness. **Please refer to Appendix D.7.** The empirical results confirm that the theoretical lower bound tracks the growth of the empirical risk in the correct orde across the entire extraction process, thereby supporting the tightness guarantees established in Proposition 1 and Theorem 2.
>
> ### Response to W4 (Graph Dataset & Computational Costs)
> Thank you for this insightful feedback. Under the no-training-data assumption, our graph generator is initialized from a lightweight GraphVAE prior (followed the initialization in [1]) and refined solely through query feedback, which aligns with the “data-free” setting. Regarding computational cost, TRIDENT indeed performs more local optimization steps than baseline attacks due to its generator–surrogate interplay. However, all components are intentionally lightweight. More importantly, in real MLaaS environments, the dominant cost is the query budget. API calls are rate-limited, often monetized, and may incur seconds of network latency, while local GPU/CPU computation for the attacker is essentially free. Our design focuses on this regime, where the attacker’s primary goal is to achieve the strongest possible extraction under limited query access, rather than minimizing local training time. **We have added a runtime table and explicit complexity discussion in Appendix D.6 to clarify this trade-off. Please refer to Appendix D.6.** Our empirical results and complexity analysis show that TRIDENT introduces moderate local computation while yielding substantially better extraction under the same query budget, supporting our claim that local optimization is a worthwhile trade-off in realistic MLaaS threat models.
>
> ### Response to W5 (Related Works)
> Thank you for your valuable feedback. We appreciate the pointer to [2], it is indeed relevant as an important security risk in graph learning. **We've included it in the related-work section in Appendix D.1.**
>
> [1] Zhuang, Yuanxin, et al. "Unveiling the Secrets without Data: Can Graph Neural Networks Be Exploited through {Data-Free} Model Extraction Attacks?." *33rd USENIX Security Symposium (USENIX Security 24). 2024*.
> [2] Huang, Renhong, et al. "Extracting training data from molecular pre-trained models." *Advances in Neural Information Processing Systems 37 (2024): 97948-97971.*

---

> > ### Comment · Reviewer_M3Vq · 2025-11-25
> >
> > Thank you for author's detailed reply. My questions, such as those regarding initialization as well as empirical study, have been resolved basically, and I am willing to revise my review score.

---

> ### Author Response · Authors · 2025-11-25
> **Additional experimental results (1/1)**
>
> We thank you again for your valuable comments and updating the score. If you have any remaining questions or concerns, please don't hesitate to share them with us, and we will be happy to respond. Below are our additional experimental results.
> ### Response to W4: Computational Costs - Appendix D.6
> | Model | Dataset | Attack | Mean (s) | Std (s) |
> | :--- | :--- | :--- | :--- | :--- |
> | GCN | Citeseer | 1 | 9.98 | 1.25 |
> | GCN | Citeseer | 2 | 64.58 | 0.71 |
> | GCN | Citeseer | 3 | 17.64 | 0.95 |
> | GCN | Citeseer | Trident | 146.47 | 36.68 |
> | GCN | A-Computers | 1 | 12.32 | 0.92 |
> | GCN | A-Computers | 2 | 26.73 | 0.78 |
> | GCN | A-Computers | 3 | 15.75 | 0.92 |
> | GCN | A-Computers | Trident | 138.19 | 58.21 |
> | GCN | Cora | 1 | 5.64 | 0.78 |
> | GCN | Cora | 2 | 30.23 | 1.29 |
> | GCN | Cora | 3 | 9.66 | 0.12 |
> | GCN | Cora | Trident | 98.09 | 17.09 |
> | GCN | Ogb-Arxiv | 1 | 6.75 | 3.93 |
> | GCN | Ogb-Arxiv | 2 | 37.61 | 0.16 |
> | GCN | Ogb-Arxiv | 3 | 17.44 | 0.00 |
> | GCN | Ogb-Arxiv | Trident | 158.39 | 41.88 |
> | GCN | Pubmed | 1 | 20.79 | 0.52 |
> | GCN | Pubmed | 2 | 30.94 | 0.41 |
> | GCN | Pubmed | 3 | 23.58 | 1.16 |
> | GCN | Pubmed | Trident | 102.64 | 0.56 |
> Note: "1, 2, 3" denote three baseline attacks

---

### Official Review · Reviewer_2PJS · 2025-10-31

**Soundness:** 2
**Presentation:** 2
**Contribution:** 2
**Rating:** 4
**Confidence:** 2

**Summary:**

The work investigates the subject of Model Extraction attacks, which consists of stealing a proprietary model by leveraging only query access, and specifically they focus on the field of Graph Neural Networks (GNNs).
The main contribution of the paper is to try to bridge some absent theoretical investigation that is missing from the literature, which focused mainly on empirically understanding and approaching this problem. Based on theoretical insights, the authors provided TRIDENT, a new attack strategy to better schedule the queries in the specific context of tight attack budget.

**Strengths:**

- The general problem that is considered is very interesting and aiming for a theoretical analysis is rather interesting and clearly bridges some gaps within the literature.
- The overall proposed theoretical analysis is interesting. While I have some questions regarding the computed bounds, it is still a good approach to extract insights.
- The experimental results seems to be validating the worth of the proposed method.

**Weaknesses:**

- The problem formulation section is confusing - specifically, while I like the overall definition and introduction to the problem setting, I feel that it is rather adapted to a general aspect and not specific aspect of graphs. For instance, the authors refer to $\mathcal{X} \in \mathbb{R}^n$, but in the case of graphs, we have the adjacency space and the node feature space - and therefore an adaptation of this section could be of great value. Additionally, how is the graph data distribution defined in this case?
- Proposition 1 is not very clear. While I get the main idea of trying to bound the risk, I don’t understand the statement and specifically the tightness of the low and upper-bounds constants. I have checked the proof to further understand, and while in the case of some losses, this bound seems to be small, in some cases it’s just degenerate. This remark also extends to Theorem 1.
- Some elements regarding the proposed methodology are not very clear.
    - For instance this claim: “We expect that even with less queries for each generated graph, the generator would generate higher quality graphs due to more iterations.” — What is the main fundamental and thinking around it? I find it very hard to believe that actually with les queries, the generator would still perform an “acceptable” (based on some semantic/generation metric) generation. This latter point can be seen experimentally your experimental setting, namely Figure 3.
    - From my understanding, you are interested in the problem of black-box attack, and at what point, you refer to white-box attack, which is a bit confusing when reading the paper.
    - In the experimental setting, a lot of details are not very clear. Specifically:
        - How are the target and surrogate model chosen - do they have similar architecture (number of layers, hidden dimensions …)? If it’s the case, how does this relate to the black-box assumption?
        - I couldn’t find details about the generator model, and also ablation study on how such choice affects the performance.
        - In the paper, you claim results for OGB-Arxiv, but I couldn’t find them, am I missing something?

**Questions:**

- Could you better clarify Proposition 1 and Theorem 1 and the main claim and specifically try to better illustrate the considered constants in the upper-bounds?
- Could you provide elements regarding the tightness of the provided bounds as it seems in that some cases they are vacuous and would argue worthless.
-  Could you reformulate the methodology section such as to clarify where does the black-box and white-box paradigm occurs?
- Could you add details regarding the experimental details (please refer to the specific questions above).

---

> ### Author Response · Authors · 2025-11-24
>
> Thank you for your insightful feedback. We sincerely appreciate your acknowledgment of our proposal's insights. Our detailed response to your concerns is as follows:
>
> ### Response to W1 (Problem Formulation)
> Thanks for pointing this out. We agree that making the connection to graph-structured inputs more explicit would improve clarity. We introduce $\mathcal{X}\in\mathbb{R}^n$ as a generic measurable input space, following standard learning-theoretic conventions. This choice is not intended to restrict inputs to vectors, but to indicate that the theoretical results require only a measurable space. For graph $G=(A,X)$, where $A$ is an adjacency matrix in a discrete measurable space $\mathcal A$, and $X$ is a node-feature matrix $\mathbb{R}^{N\times d}$. The resulting input space is the product measurable space $\mathcal X=\mathcal A\times\mathcal{X}_{\text{feature}}$. **To improve clarity, we've instantiated $\mathcal X$ in the graph setting and emphasized that our general formulation directly subsumes attributed graphs.** Regarding the graph data distribution, the target distribution $T$ is a probability measure over the above product measurable space.
>
> ### Response to W2, Q1, Q2 (Proposition 1 & Theorem 1)
> Thank you for raising this point. We clarify that Proposition 1 analogous to norm equivalence in Riemannian geometry. The reviewer’s observation regarding “degenerate” cases is correct when these assumptions are not enforced. However, as detailed in Appendix A, our analysis explicitly assumes several conditions commonly adopted in deep-learning theory, such as bounded inputs, bounded hypotheses, etc. Under these assumptions, the constants remain finite, and the inequality in Proposition 1 is tight in the usual sense. **We've added an experiment with discussion in Appendix D.7 to show tightness.** Our empirical findings clearly confirm the theoretical predictions: the empirical risk increases consistently with the distribution gap, while the linear lower bound stays below it and preserves the correct growth order.
>
> ### Response to W3, Q3 (Clarification for Methodology )
> Thank you for your valuable feedback. Our claim “fewer queries per iteration but more iterations improve generator quality” follows directly from Theorem 2, increasing $\alpha$ yields finer partitions $w_i$ and therefore more informative updates. It can provide more informative gradient updates even if each iteration uses fewer queries. Importantly, the total query budget does not decrease. Fig. 3 may give the impression of fewer queries per graph due to visualization choices. Because of sparsity of the graph, we can still have acceptable graphs for each iteration. Regarding the white-box attack, we apologize for the possible misleading from Fig.1. Our intention is to help readers better understand the relationship between the light surrogate and the surrogate. As the attacker builds a light surrogate and has the whole access to it, the process of evaluating light surrogate is similar with the white-box attack. **We've revised relevant descriptions in the main text in Section 4.2.**
>
> ### Response to W3, Q4 (Experimental Details)
> Thanks for your valuable feedback. We follow the widely acknowledged black-box assumption as prior data-free MEA work [1]. The victim architecture is unknown to the attacker. The surrogate uses standard GNN backbones but never matches hyperparameters like layers or hidden dimensions of the victim. For the generator achitecture, it is a GraphVAE-style model consisting of an node-feature generator, a latent sampler, and an adjacency decoder (in Section 4.1). Because the generator cannot be removed from the pipeline, we use random graphs as an ablated version to examine its role. As for OGB-Arxiv, these results appeared in the supplementary materials, and we have now moved them to the Appendix. **Please refer to Table 9.** We note that given the relatively low accuracy of the victim model on OGB-Arxiv, our attack achieves correspondingly high fidelity in this setting.
>
> [1] Zhuang, Yuanxin, et al. "Unveiling the Secrets without Data: Can Graph Neural Networks Be Exploited through {Data-Free} Model Extraction Attacks?." *33rd USENIX Security Symposium (USENIX Security 24). 2024*.

---

> ### Author Response · Authors · 2025-11-25
> **Additional experimental results (1/1)**
>
> We thank you again for your valuable comments. Below are our additional experimental results.
> ### Response to Q4: Missing OGB-Arxiv Results - Appendix Table 9
> | Model | Method | A-Photos (Accuracy) | A-Photos (Fidelity) | OGB-Arxiv (Accuracy) | OGB-Arxiv (Fidelity) |
> | :--- | :--- | :--- | :--- | :--- | :--- |
> | **GAT** | Real Data | 17.70 ± 6.97 | 23.02 ± 4.94 | 10.82 ± 1.21 | 39.68 ± 2.76 |
> | | Random Graph | 11.21 ± 3.56 | 10.23 ± 2.89 | 8.72 ± 0.71 | 21.17 ± 2.64 |
> | | Attack III-A | 61.72 ± 1.24 | 83.75 ± 3.00 | 14.73 ± 0.53 | 78.62 ± 0.47 |
> | | Attack III-F | 63.15 ± 2.23 | 88.61 ± 2.15 | 13.71 ± 0.37 | 77.31 ± 0.64 |
> | | TRIDENT | **72.78 ± 0.20** | **90.32 ± 0.81** | **30.29 ± 0.19** | **99.23 ± 0.27** |
> | **GCN** | Real Data | 18.92 ± 4.53 | 24.29 ± 2.07 | 9.17 ± 0.72 | 20.49 ± 1.29 |
> | | Random Graph | 13.41 ± 2.71 | 10.57 ± 1.83 | 7.41 ± 0.21 | 12.33 ± 1.74 |
> | | Attack III-A | 54.31 ± 0.37 | 89.93 ± 1.72 | 15.31 ± 0.59 | 68.29 ± 3.82 |
> | | Attack III-F | 57.21 ± 2.45 | 87.14 ± 2.42 | 17.28 ± 0.42 | 72.51 ± 3.71 |
> | | TRIDENT | **72.80 ± 0.60** | **88.73 ± 0.83** | **29.41 ± 0.52** | **99.23 ± 0.24** |
> | **SAGE** | Real Data | 23.15 ± 3.42 | 23.23 ± 5.66 | 10.85 ± 1.17 | 18.62 ± 2.79 |
> | | Random Graph | 11.89 ± 3.75 | 18.20 ± 4.12 | 7.51 ± 0.69 | 16.75 ± 3.12 |
> | | Attack III-A | 63.82 ± 5.79 | 80.80 ± 6.65 | 8.72 ± 0.61 | 55.21 ± 1.43 |
> | | Attack III-F | 59.21 ± 3.85 | 85.79 ± 4.11 | 10.42 ± 0.81 | 53.53 ± 1.25 |
> | | TRIDENT | **75.92 ± 0.68** | **87.52 ± 2.21** | **27.37 ± 0.33** | **99.23 ± 0.21** |

---

> > ### Comment · Reviewer_2PJS · 2025-11-27
> >
> > I thank the authors for their great rebuttal that answered the majority of my previous concerns. I think the manuscript looks a little better for the moment from my understanding, and therefore I would increase my score to the positive side.
> >
> > Thank you again for taking into account my questions and suggestions.

---

> > > ### Author Response · Authors · 2025-11-27
> > >
> > > Thank you for raising our scores and for the helpful suggestion. If any concerns remain, we would be happy to address them.

---

### Official Review · Reviewer_94Zi · 2025-11-01

**Soundness:** 3
**Presentation:** 3
**Contribution:** 3
**Rating:** 6
**Confidence:** 3

**Summary:**

The core contribution is a new theoretical framework that formalizes extraction risk. The authors derive a lower bound for this risk, tying it directly to the victim model's own risk and, most importantly, the generalization discrepancy between the attacker's query distribution and the victim's (unseen) target distribution . This yields a clear design principle: an effective attack must minimize this discrepancy.

**Strengths:**

The paper's primary strength is its shift from a purely empirical to a theory-driven attack design. Formalizing the problem in terms of extraction risk and generalization discrepancy provides a solid conceptual foundation. The budget ratio is a simple but highly effective mechanism to manage the trade-off . The ablation studies  clearly show this component significantly boosts both performance and efficiency . The attack is thoroughly evaluated. The method shows consistent SOTA performance across different datasets  and different victim architectures. The attack also remains effective even against a label-flipping defense . The budget ratio is a simple but highly effective mechanism to manage the query-budget-to-training-iteration trade-off. The ablation studies (Fig. 3) clearly show this component significantly boosts both performance and efficiency .

**Weaknesses:**

The paper is vague on the specifics of the light surrogate model. It is described as smaller and can be "initialized from" the main surrogate.

**Questions:**

is there a principled way to pick $\alpha$ given a known budget B?

What prior knowledge is assumed for this "data-free" attack?

---

> ### Author Response · Authors · 2025-11-24
>
> Thank you for your valuable feedback. We sincerely appreciate your acknowledgment of our proposal's insights. Our detailed response to your concerns is as follows:
>
> ### Response to W1 (Light Surrogate Model)
> We apologize for the insufficient detail. The main purpose of the light surrogate is to provide a low-capacity approximation of the main surrogate so that their disagreement highlights informative regions. We note that the light surrogate shares the same backbone (GCN/GAT/SAGE) as the main surrogate to ensure architectural consistency. However, its capacity is intentionally restricted by fewer layers and smaller hidden dimensions. Then it can correctly predict “easy” samples while failing on harder ones, creating the desired selective pressure for the main surrogate. Regarding initialization, only the final classifier layer of the light surrogate is re-initialized at each refinement step, whereas other layers inherit parameters from the main surrogate. This ensures both rapid convergence of the KL term and stable evaluation of disagreement without extra queries. **We've refined relevant descriptions in the main text in Section 4.2.**
>
> ### Response to Q1 ($\alpha$ for Given buget)
> Thank you for the insightful question. The purpose of the budget ratio $\alpha$ is to determine how many refinement updates TRIDENT can perform under the same total query budget. Theorem 2 and Proposition 2 imply that more and smaller update steps help reduce the discrepancy term by enabling finer-grained partitions $w_i$. Thus, under a fixed budget $B$, our goal is $\max t=\max(\alpha t_0)$ such that $\sum_{j=1}^{\alpha t}\frac{|G_j|}{\alpha}\approx B$. Empirically, after generator warm-up, $|G_j|$ stays roughly stable, allowing to choose $\alpha$ approximately as [desired # iterations (refinement)]/[original # iterations (baseline)]. Thus, we recommend $\alpha\in[10,20]$ when $B$ is medium to large, $\alpha\in[5,9]$ for low budgets, and $\alpha=1$ only when generator updates are expensive, e.g., very large graphs.
>
> ### Response to Q2 (Prior Knowledge of data-free setting)
> Thanks for your valuable feedback. We strictly follow the prior-knowledge assumptions defined in the data-free GNN MEA setting introduced by STEALGNN [1], which is the most restrictive threat model in current literature. Our attack does not assume any additional knowledge beyond this standard setting. Specifically, we assume that (1) Black-box query access only. The attacker can only query the victim API and obtain its predictions (hard or soft labels). (2) No access to the victim’s training data, such as node attributes, graph topology, etc. We consist with [1] to assume that the attacker only knows the expected input dimension of the victim model, and that the victim model processes attributed graphs. It is unavoidable and is the minimal assumption for API-based services.
>
> [1] Zhuang, Yuanxin, et al. "Unveiling the Secrets without Data: Can Graph Neural Networks Be Exploited through {Data-Free} Model Extraction Attacks?." *33rd USENIX Security Symposium (USENIX Security 24). 2024*.

---

> ### Author Response · Authors · 2025-11-25
>
> We thank you again for your valuable comments. If you have any remaining questions or concerns, please don't hesitate to share them with us, and we will be happy to respond.

---

### Author Response · Authors · 2025-11-25
**Global Response**

Dear Reviewers and Area Chairs:

We sincerely thank all reviewers for their valuable comments and constructive suggestions. We have revised the paper accordingly, with all updates highlighted in **blue** in the revised version. Specifically, based on the feedback from Reviewers 94Zi, 2PJS, M3Vq, and Peod, we have summarized the revisions into the following key updates:

1. **Expanded Experimental Scope and Efficiency Analysis** (Reviewers M3Vq, Peod)
- We added a detailed complexity analysis and a runtime table comparing TRIDENT with baselines. **These details are provided in Appendix D.6.**
- We extended our evaluation to the link prediction task to demonstrate TRIDENT's effectiveness beyond node classification. **These results are included in Appendix D.5.**
- We added a sensitivity analysis for KL weight to address stability concerns (Reviewer Peod). **This is found in Appendix C.5.**
- We moved the OGB-Arxiv results and A-Photos results to **Table 9 in the Appendix** for completeness.

2. **Theoretical Validation and Tightness** (Reviewers 2PJS, M3Vq, Peod)
- We added an experiment visualizing the relationship between the empirical extraction risk and our derived lower bound, confirming that the bound effectively tracks the risk's growth order. **This validation is presented in Appendix D.7.**

3. **Methodological Clarifications** (Reviewers 94Zi, 2PJS, M3Vq)
- We refined the description of the "Surrogate-as-Victim" module, clarifying the capacity constraints and initialization of the light surrogate to prevent misunderstandings regarding white-box assumptions. **This revision is in Section 4.2.**
- We instantiated the input space $\mathcal{X}$ specifically for graph data and refined the definitions of extraction risk and generalization discrepancy to reduce notation redundancy. **These updates are in Section 2.**

To facilitate easy access, we have also provided these additional experimental results in tables within the corresponding individual responses. We appreciate the reviewers’ thoughtful feedback and positive recognition. Their suggestions significantly strengthened this paper, and we hope the revisions adequately address all concerns. We look forward to any further discussion!

Best,

Authors of Paper #6728

---

> ### Author Response · Authors · 2025-12-01
>
> We sincerely thank the AC for the time and effort devoted to handling our submission, especially given the heavy workload this year. The reviewers have confirmed that our responses satisfactorily resolved their questions and had raised their scores, as noted in our discussion. We believe these revisions fully address all reviewer concerns and have substantially enhanced the overall quality of the paper. We thank the AC again, and your work is highly valuable to the ICLR community!

---

### Meta-Review · Area_Chair_rHND · 2026-01-07

**Summary:**

1. **Baseline coverage remain insufficient.** Despite the added experiments, the empirical comparison still does not fully situate TRIDENT against the broader landscape of recent graph model extraction attacks and related security methods. Multiple reviewers noted that important related works and representative baselines were either missing or only briefly discussed, which weakens the paper’s positioning and makes it difficult to assess(Reviewers 2PJS, M3Vq, Peod).

2. **The writing and presentation significantly hinder clarity and accessibility.** Several reviewers consistently raised concerns that the paper is overly formal and dense, with heavy notation and stacked definitions that obscure the main intuition. Key components of the methodology (e.g., surrogate-as-victim alignment, generator behavior, and the black-box vs. white-box distinction) were initially unclear and required substantial clarification during rebuttal. Although some explanations were improved in later revisions, the overall presentation remains hard to follow, and important design choices are still not clearly motivated or explained in a reader-friendly manner (Reviewers 2PJS, M3Vq, Peod).

3. **Initial concerns regarding missing ablations, hyperparameter sensitivity, and task diversity were valid and substantial**. While the authors later added ablation studies, parameter analyses, and link prediction experiments that partially resolve these issues, these additions highlight that the original submission was experimentally underdeveloped, and the current completeness relies heavily on post-rebuttal material rather than a strong initial evaluation (Reviewers Peod, M3Vq).

4. **Scalability and computational cost remain a serious concern.** Although the authors argue that query efficiency is the primary metric in MLaaS settings, the provided runtime analysis shows that TRIDENT often incurs an 10X higher local computation time than baseline attacks. The reported results do not convincingly support the claim that the method scales well in practice, especially for larger graphs, and raise questions about whether the theoretical query efficiency translates into a practically deployable attack framework under realistic resource constraints (Reviewers Peod, M3Vq).

**Reviewer Concerns:**

Initial concerns regarding missing ablations, hyperparameter sensitivity, and task diversity were well addressed.
However, the three major concerns are still outstanding:
1. Limited coverage and discussions of related baseline methods
2. The writing and presentation quality can not be addressed in the short period of rebuttal.
3. Scalability and computational cost remain a serious concern.

**Reviewer Scores:**

I believe Reviewer 2PJS would likely maintain his negative score, since their core concerns remain largely unresolved. For Reviewers 94Zi, M3Vq, and Peod, who gave borderline-accept scores (6), I expect they would stay borderline, because the important concerns they raised were not convincingly addressed in the revisions.

---

### Decision · Program_Chairs · 2026-01-26

Reject